# Cryo-EM structures of prokaryotic ligand-gated ion channel GLIC provide insights into gating in a lipid environment

Nikhil Bharambe [1,4], Zhuowen Li [1,4], David Seiferth [2], Asha Manikkoth Balakrishna [1], Philip C. Biggin [2] & Sandip Basak [1,3] ✉

GLIC, a proton-activated prokaryotic ligand-gated ion channel, served as a model system for understanding the eukaryotic counterparts due to their structural and functional similarities. Despite extensive studies conducted on GLIC, the molecular mechanism of channel gating in the lipid environment requires further investigation. Here, we present the cryo-EM structures of nanodisc-reconstituted GLIC at neutral and acidic pH in the resolution range of 2.6 – 3.4 Å. In our apo state at pH 7.5, the extracellular domain (ECD) displays conformational variations compared to the existing apo structures. At pH 4.0, three distinct conformational states (C1, C2 and O states) are identified. The protonated structures exhibit a compacted and counter-clockwise rotated ECD compared with our apo state. A gradual widening of the pore in the TMD is observed upon reducing the pH, with the widest pore in O state, accompanied by several layers of water pentagons. The pore radius and molecular dynamics (MD) simulations suggest that the O state represents an open conductive state. We also observe state-dependent interactions between several lipids and proteins that may be involved in the regulation of channel gating. Our results provide comprehensive insights into the importance of lipids impact on gating.

Ligand-gated ion channels play crucial roles in the transmission of electrochemical signals among neurons. The eukaryotic pentameric ligand-gated ion channels (pLGICs), also known as Cys-loop receptors, are involved in rapid synaptic transmission in the peripheral and central nervous systems. These receptors are categorized based on the properties of the ions they conduct. Cation-selective family members such as serotonin receptors (5-HT$_3$Rs) and nicotinic acetylcholine receptors (nAChRs) are referred to as excitatory receptors. Anion-selective members such as glycine receptors (GlyRs) and γ-aminobutyric acid receptors (GABA$_A$Rs) are known as inhibitory receptors. Despite the diverse sequences among pLGICs, the overall architectural fold remains similar. Each receptor comprises an agonist-specific extracellular domain (ECD), a transmembrane domain (TMD) that houses a gate and selectivity filter, and a poorly understood intracellular domain (ICD). The complex gating mechanism of pLGICs[1,2] involves a cascade of conformational changes in each domain induced by agonist binding at the ECD, leading to transitions among closed, open, and desensitized states. Besides these major functional states, pLGICs can adopt various intermediate states, and the interplay among these states regulates the signal transmission in neurons[1,3]. Hence, dysregulation of channel function could lead to impaired nerve communications and contribute to neurological diseases such as Alzheimer's, Parkinson's, schizophrenia, and epilepsy, making pLGICs prime targets for developing therapeutic agents[4–6]. Although significant progress has been made in the Cys-loop receptor field[7–21], it remains a challenge to capture major and transitional conformational states in a lipid environment using traditional structural biological techniques such as X-ray crystallography or NMR. The recent

[1]School of Biological Sciences, Nanyang Technological University, Singapore 637551, Singapore. [2]Structural Bioinformatics and Computational Biochemistry, Department of Biochemistry, University of Oxford, Oxford, UK. [3]NTU Institute of Structural Biology, Nanyang Technological University, Singapore 639798, Singapore. [4]These authors contributed equally: Nikhil Bharambe, Zhuowen Li. ✉e-mail: sandip.basak@ntu.edu.sg

advancement of single particle cryo-electron microscopy (cryo-EM) has facilitated the characterization of ion channels in multiple conformational states in near-native environments[12,13,18,22–27]. However, questions persist regarding the unequivocal assignment of functional states to specific conformations, particularly in prokaryotic ligand-gated ion channels. Given the substantial modulation of pLGICs by membrane lipids[28–36], leveraging cryo-EM to determine high-resolution structures in a lipid environment would not only validate detergent-solubilized structures, but also enhance our understanding of the influence of lipids on channel gating. This, in turn, facilitates the assignment of conformational states to their closest physiological states.

GLIC, a prokaryotic ligand-gated ion channel found in *Gloeobacter violaceus*, served as a model system to comprehend the gating mechanism of eukaryotic ligand-gated channels. The channel is activated at low pH, and the residues responsible for proton sensing[37,38] are located in ECD but away from the canonical orthosteric binding site. The $pH_{50}$ of liposome-reconstituted GLIC is ~2.9[39]; however, the measurement by two-electrode voltage clamp (TEVC) experiments on *Xenopus* oocytes yielded a $pH_{50}$ of 5.5[40,41]. This discrepancy underscores the influence of the membrane environment on the channel gating. Despite the intrinsically missing ICD in GLIC, upon activation, the channel undergoes conformational changes reminiscent of eukaryotic orthologs[9]. Notably, GLIC contains conserved cavities recognized by general anesthetics, local anesthetics, alcohols, and channel blockers[42–44]. However, all the deposited GLIC structures were obtained in pure or partial detergent environments, except one that was solved at low resolution using the meso crystallization technique[7–10,45–47]. Despite the structures being solved in detergent, few lipid molecules were found to interact with the transmembrane domain of the channel, which were believed to be extracted during solubilization or introduced by the addition of *E. coli* polar lipid extract during crystallization[10,11,42]. Since lipids have been shown to modulate the conformational dynamics of pLGICs[13,48], the effect of detergents on the conformational states of GLIC remains a topic of discussion[28]. Existing GLIC structures demonstrated potential resting or closed state, locally closed states induced by mutations or chemical cross-linking, pre-open states, putative open, and putative desensitized states[1,7,9,10,49,50]. Notably, in the putative open state, several detergent molecules occlude the pore, posing a non-physiological scenario[7,8]. The effect of trapped detergent molecules on the pore radius and the precise assignment of physiological open state(s) is a subject of further investigation[51]. Recently, studies on GLIC mainly focused on either identifying the residues responsible for proton-sensing or gaining a better understanding of the effect of modulators[37,38,50,52]. A recent structural investigation of detergent-solubilized GLIC by the cryo-EM at various pH conditions (pH 7.0, 5.0, and 3.0) captured closed states distinct from known GLIC structures[46]. Contrary to the existing knowledge of large twisting and compaction in the ECD upon activation[53], the conformational changes in the ECD and TMD induced by protonation are subtle with an overall backbone (Cα) RMSD ~0.5 Å[28,54]. The structure of GLIC at pH 3.0 reconstructed from the minor populations of protein particles displays rotation, compaction of the ECD, and partial expansion of the pore. Similar conformational changes were observed in the minor populations of both pH 7.0 and pH 5.0 structures, possibly due to spontaneous opening previously observed in unliganded pLGICs[55,56]. On a cautionary note, the poor resolutions (~5 Å) of the maps reconstructed from the minor populations of the protein particles might have yielded less accurate models that could influence the structural comparisons. The absence of lipid density in these structures raises questions about the crucial role of lipids in stabilizing functional states of GLIC and modulating conformational flexibility upon activation. To understand the effect of membrane lipids on channel gating, structural investigation of GLIC in detergent-free lipid environments is of utmost importance.

In this study, we present cryo-EM structures of GLIC reconstituted in lipid nanodiscs at pH 7.5, 5.5, 4.0, and 2.5. We identified a resting-like state, two putative intermediate states, a putative pre-open state, and an open state. Our structures show a cascade of potential conformational changes in both ECD and TMD upon pH reduction. The open state possesses an unobstructed ion permeation pathway with several layers of water molecules at the pore. Molecular dynamics (MD) simulations confirm a hydrated and $Na^+$ ion-permeable pore. A number of lipid densities are observed near TMD in all of our structures. While the position and interaction patterns of lipids at the lower leaflet are conserved, lipids at the upper leaflet show dynamic repositioning with notable variation in their interactions across these states, indicating a state-dependent role, as previously predicted in other pLGICs using MD simulations[48]. Together with MD simulations data, our cryo-EM structures of nanodisc-reconstituted GLIC provide better insights on the gating mechanism of GLIC in a lipid environment.

## Results

### A distinct structure of nanodisc-reconstituted GLIC in the apo state compared to the existing detergent-solubilized apo state

Although X-ray crystallography has provided substantial structural information on GLIC, the challenges persist in obtaining diffraction-quality crystals in a detergent-free lipid environment. Most crystal structures of GLIC feature low lipid occupancy at the TMD with three confidently identified lipid binding sites in each subunit[7], which requires further investigation on the conformational stability of GLIC in lipid environments. To investigate the role of lipids on channel gating, GLIC was initially purified in detergent and subsequently reconstituted in lipid nanodiscs using the scaffolding protein MSP1E3D1. Two cryo-EM structures of GLIC reconstituted in either soybean polar lipid extract (or asolectin referred to as "aso" onwards) or DOPE:POPS:POPC (2:1:1) at pH 7.5 were obtained with a nominal resolution of 3.4 and 2.9 Å, respectively (Supplementary Figs. 1, 2a, c, f, h). Local resolution estimation of these two maps (referred here onwards as $GLIC_{aso-pH7.5}$ and $GLIC_{pH7.5}$) in the apo state shows higher resolution in TMD than ECD (Supplementary Fig. 2b, g). The final maps exhibit clear density for both ECD and TMD and were used for model building (Supplementary Fig. 2). The overall architecture of GLIC is similar to the existing structures with each monomer featuring two domains: the ECD, mainly composed of β strands and the TMD, composed of four α helices (M1-M4) (Fig. 1a). Several lipid molecules with clear polar head groups were modeled onto non-protein densities which were identified near the TMD (Supplementary Fig. 2a, f). A comparison between $GLIC_{aso-pH7.5}$ and $GLIC_{pH7.5}$ structures shows overlapping secondary structures with minimum conformational differences and a backbone (Cα) RMSD of 0.52 Å (Fig. 1b and supplementary Fig. 3a). Subtle conformational changes in the upper portion of the ECD, particularly at the loop regions, suggest that both lipid mixtures stabilize GLIC in a similar conformational state at neutral pH (Fig. 1c). A comparison of our $GLIC_{pH7.5}$ structure with the existing structure of GLIC in resting/apo state ($GLIC_{detergent-pH7.5}$; PDB: 4NPQ) revealed notable conformational differences. In contrast to $GLIC_{detergent-pH7.5}$, the ECD in our apo state is twisted clockwise (Fig. 1d) along with the outward movements of Loop F and Loop C from the central axis (Fig. 1e, f). Similar outward motion is observed in all the β strands of our apo state, accompanied by a large clockwise twist (-15°) in the β1–β2 loop (Fig. 1g). Moreover, Loop A moves upward in our apo state with a maximum Cα distance of ~3.1 Å and twisted ~8° clockwise (Fig. 1g). In addition to rotation, an overall expansion of the ECD is observed in our apo state compared to $GLIC_{detergent-pH7.5}$ structure, which is reflected by increased accessible surface area (ASA) from 48,123 to 54,089 Å² (V5-R192) (Supplementary Fig. 3c, d). The rotation and expansion results in the backbone (Cα) RMSD in ECD of 2.7 Å (Supplementary Fig. 3b). Despite high conformational similarities in the TMD (Fig. 1d), the M2–M3 loop appeared to move upward toward

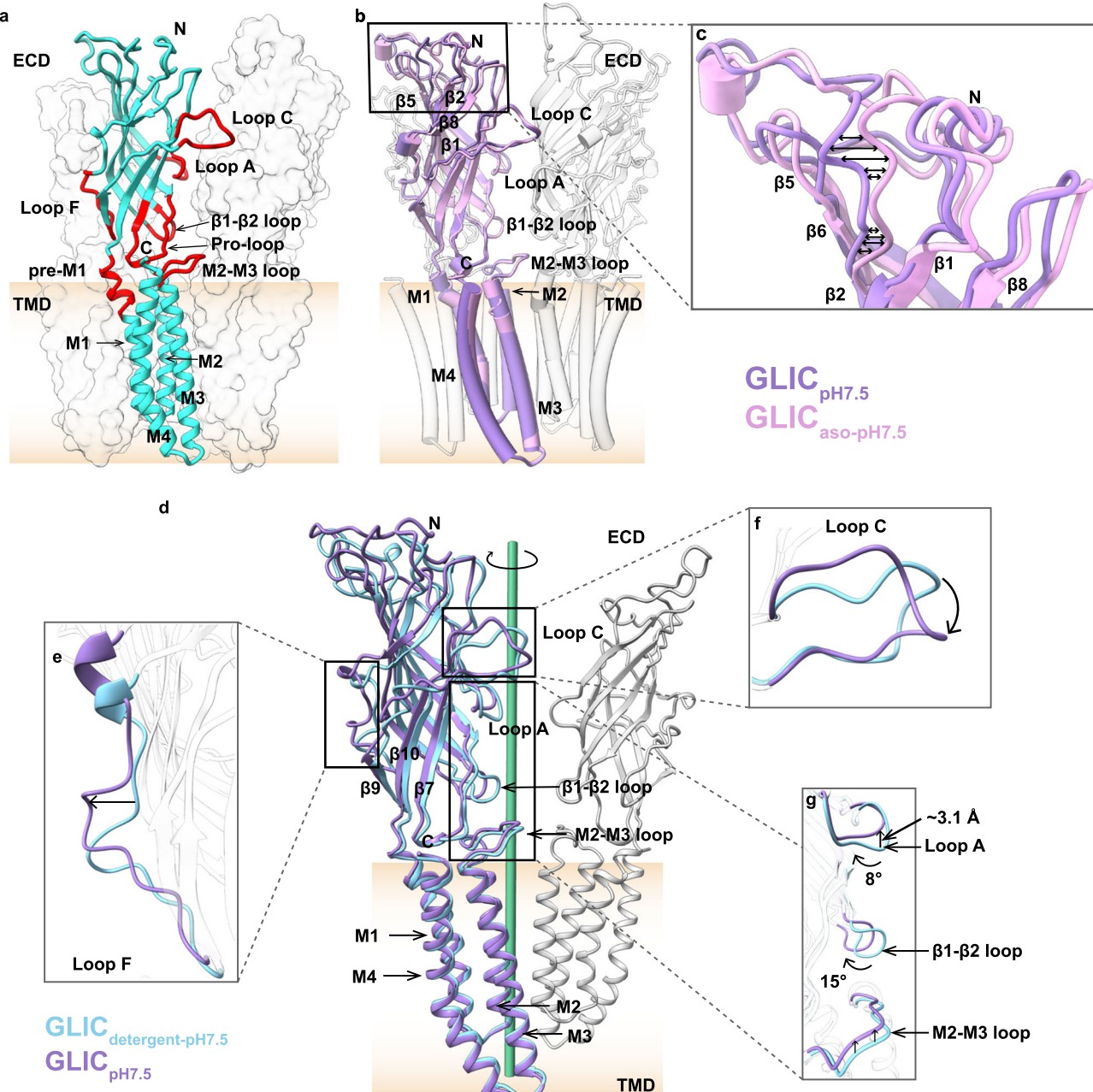

**Fig. 1 | Comparison of the crystal and cryo-EM structures of GLIC in the apo state. a** Overall structure of GLIC in a cartoon representation. One subunit is colored cyan for clarity. Important regions of the model are colored red. The membrane bilayer is shown as a gradient of apricot color. **b** Superposition of GLIC$_{aso-pH7.5}$ (plum) and GLIC$_{pH7.5}$ (purple) structures by TMD in a cartoon representation. **c** The upper part of the ECD shows subtle changes in the loops indicated by the bidirectional arrows. Helices are shown as cylinders. **d** Superposition of GLIC$_{pH7.5}$ (purple) and GLIC$_{detergent-pH7.5}$ (sky blue, PDB ID: 4NPQ) by TMD shows conformational differences in Loop F (**e**), Loop C (**f**), Loop A (**g**), β1-β2 loop (**g**), and M2−M3 loop (**g**), which are highlighted in zoom view. Only one subunit is colored for clarity. The central axis is shown as a green cylinder. Arrows indicate the direction of movement.

the ECD in our apo structure (Fig. 1g). Similar overall conformational differences are observed in our apo state when compared with the cryo-EM structure of detergent-solubilized GLIC at pH 7.0 (GLIC$_{detergent-pH7.0-cryo-EM}$) exhibiting a backbone (Cα) RMSD in ECD of 3.5 Å (Supplementary Fig. 3e)[46]. In summary, our apo state manifests a distinct conformation, featuring an expanded and clockwise twisted ECD compared to the existing apo states.

**The structures of nanodisc-reconstituted GLIC in the protonated states adopt conformations distinct from our apo state**

To investigate protonated conformations in the lipid environment, we studied nanodisc-reconstituted GLIC at pH 5.5 and 4.0. A severe

aggregation of GLIC reconstituted in asolectin was observed on cryo-EM grids and posed challenges in data collection. Therefore, GLIC was reconstituted in nanodiscs using DOPE:POPS:POPC lipid mixture, which stabilized the protein on cryo-EM grids at low pH. Approximately 6600 and 14,000 movies were collected at pH 5.5 and 4.0, respectively. Iterative data processing using CryoSPARC[57] and RELION[58] programs yielded the density map of GLIC at pH 5.5 (GLIC$_{pH5.5}$) with a nominal resolution of 2.7 Å (Supplementary Figs. 4, 5). The overall traceable densities of the GLIC$_{pH5.5}$ map facilitated building a full-length model (Supplementary Fig. 5a). Extensive data processing of GLIC at pH 4.0 separated three major 3D classes, resulting in three distinct refined maps with nominal resolutions of 2.7,

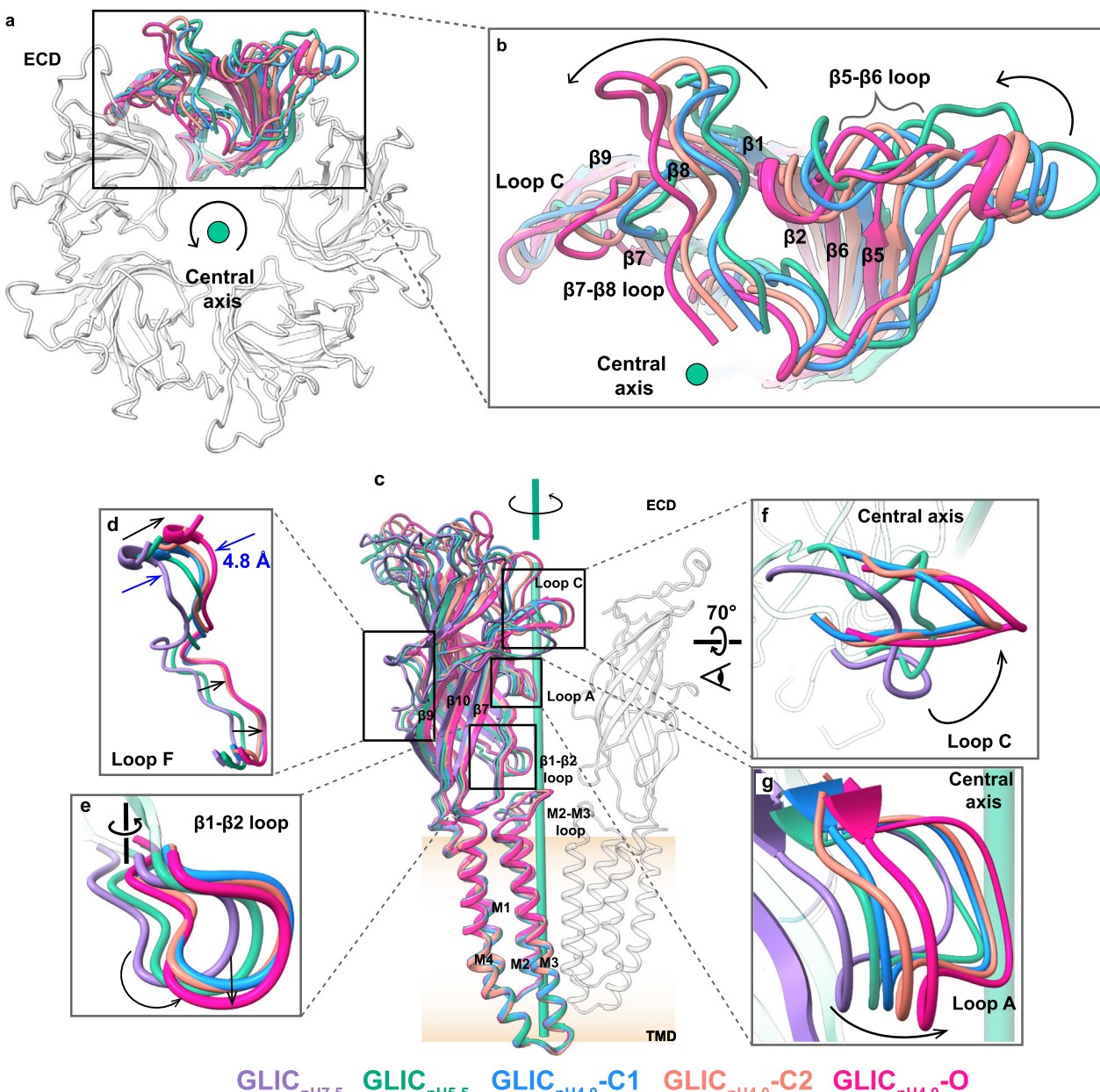

**Fig. 2 | Comparison of ECD in the apo and protonated states. a** Top view of the ECD shows gradual counter-clockwise rotation in the protonated states. All models were aligned with TMD. For clarity, only one subunit from each structure is colored as follows: GLIC$_{pH5.5}$ (green), GLIC$_{pH4.0}$-C1 (dodger blue), GLIC$_{pH4.0}$-C2 (salmon), and GLIC$_{pH4.0}$-O (deep pink). Other subunits from a representative structure are in gray. **b** A zoom-in view of the ECD from a single subunit is shown. Arrows indicate counter-clockwise movement. The green circle represents the central axis. **c** Overall comparison of GLIC in the apo and protonated states. Only one subunit is colored according to the individual structure and a diagonal subunit is in gray. Zoom-in views of various regions show the conformational changes in Loop F (**d**), β1-β2 loop (**e**), Loop C (**f**), and Loop A (**g**) from apo state to protonated states. Arrows indicate the key direction of conformational changes.

3.3, and 2.7 Å which correspond to three states, namely C1 or GLIC$_{pH4.0}$-C1, C2 or GLIC$_{pH4.0}$-C2, and O or GLIC$_{pH4.0}$-O, respectively (Supplementary Figs. 6–9). The TMD manifests higher local resolution than the ECD, suggesting a less flexible and more defined conformation in the lipid environment (Supplementary Figs. 5b, 7–9b). While the proton sensors have been identified previously[38,53], the current resolution of the maps did not allow the modeling of protons. Nevertheless, conformational changes upon activation are clearly evident in the protonated structures. Compared to the GLIC$_{pH7.5}$ state, a gradual counter-clockwise rotation in ECD around the central axis is observed in the order of GLIC$_{pH5.5}$, GLIC$_{pH4.0}$-C1, GLIC$_{pH4.0}$-C2, and GLIC$_{pH4.0}$-O structures, and the backbone (Cα) RMSD of ECD is calculated to be 2.9, 3.3, 3.3, and 3.6 Å, respectively (Fig. 2a–g and supplementary Fig. 10).

The concerted twist in the ECD led to the repositioning of loops at the interface (Fig. 2c). Loop F in all the structures at pH 4.0 adopted a similar conformation by moving inward toward the central axis compared to the apo state (Fig. 2d). A gradual counter-clockwise rotation is prominent in the β1-β2 loop at lower pH, with a maximum rotation of ~16° observed in the O state compared to the apo state (Fig. 2e). At pH 5.5, Loop F and β1-β2 loop are positioned between the apo and pH 4.0 states (Fig. 2d, e). Loop C gradually moved inward toward the central axis and adopted closed conformations at all low pH states, with the minimum and maximum (-11°) counter-clockwise rotation observed in GLIC$_{pH5.5}$ and GLIC$_{pH4.0}$-O states, respectively (Fig. 2f). Loop C in both C1 and C2 states adopts a similar conformation, which lies between the GLIC$_{pH5.5}$ and O states (Fig. 2f). A counter-clockwise

rotation is observed in Loop A when the pH is reduced, with the highest rotation of ~16° (Fig. 2g). Similar counter-clockwise rotations are observed in the β strands located at the outer surface of ECD. The rotational rearrangement of the loops and β strands at the subunit interface, particularly in GLIC$_{pH4.0}$-O, causes several polar interactions (Supplementary Fig. 11a). The guanidine group of R77 located at the β5 strand of the principal (+) subunit interacts with D88 located at the β5 strand of complementary (−) subunit (Supplementary Fig. 11a). A complex intersubunit interaction network is observed involving residues in Loop A. The backbone carbonyl oxygen of F78 and V79 located at Loop A of the principal (+) subunit interacts with the guanidine group of R105 located at the β6 strand of complementary (−) subunit (Supplementary Fig. 11a). Similarly, V81 from the principal (+) subunit interacts with N40 from the complementary (−) subunit (Supplementary Fig. 11a). The backbone carbonyl atom of E82 located at Loop A of the principal (+) subunit interacts with Y28, N40, and S107 from the complementary (−) subunit (Supplementary Fig. 11a). In addition, D136 located at β7-β8 loop of the principal (+) subunit interacts with T65 of the complementary (−) subunit (Supplementary Fig. 11b). Due to the counter-clockwise rotation, D178 and R179 located in Loop C of the principal (+) subunit interact with K148 and D91 of the complementary (−) subunit, respectively (Supplementary Fig. 11b). Similar interaction networks were also observed in the existing putative open state (GLIC$_{detergent-pH4.0}$-O). However, these interactions are not observed in GLIC$_{pH5.5}$, C1, and C2 structures because of the lesser extent of twist in the ECD compared to GLIC$_{pH4.0}$-O. In addition to notable intersubunit interactions at the ECD, an intra-subunit triad is established in GLIC$_{pH4.0}$-O due to a subtle counter-clockwise rotation and downward movement of β1-β2 loop toward TMD (Fig. 2e). This triad formed by the interaction of R192 located in the pre-M1 with D32 in the β1-β2 loop close to the key proton sensor E35, and D122 in Pro-loop has been demonstrated to play a crucial role in signal transduction at the ECD-TMD interface and is critical for channel function[53,59] (Supplementary Fig. 11c). Notably, a similar triad is also observed in our C2 state due to the subtle rotation of β1-β2 loop (Fig. 2e). In addition to the rotation, a compaction of the ECD occurs when the pH is reduced from 7.5 to 4.0 (Fig. 3a, b and Supplementary Movie 1). Due to the rotation and compaction, the accessible surface area (ASA) of the ECD (V5-R192) in GLIC$_{pH4.0}$-O (47,499 Å²) reduced compared to GLIC$_{pH7.5}$ (54,089 Å²). An increase in buried surface area (BSA) is observed in the ECD of GLIC$_{pH4.0}$-O (7683 Å²) compared to GLIC$_{pH7.5}$ (2608 Å²), supporting the reduced ASA. The inward movement of Loop F with a maximum backbone displacement ~4.8 Å (at V149) toward the central axis in low pH states also contributes to their compaction (Fig. 2d). To further investigate the conformational changes of GLIC in a more acidic environment, we determined the cryo-EM structure at pH 2.5. Iterative 3D classifications and refinements of the data yielded the final map with a nominal resolution of 2.6 Å (Supplementary Figs. 12, 13). Surprisingly, the structure of GLIC at pH 2.5 (GLIC$_{pH2.5}$-O) resembles the GLIC$_{pH4.0}$-O with an overall backbone (Cα) RMSD of 0.20 Å (Supplementary Fig. 13f).

Conformational changes in the TMD of the protonated states are not as prominent as those observed in the ECD, but a pattern of concerted movements of M2 upon protonation is clearly visible (Supplementary Movie 2). Upon reducing the pH to 5.5, a subtle counter-clockwise rotation is observed at the TMD compared to the apo state (Fig. 3c), with the maximum twist observed at the upper part of M2, which in turn moves the beginning of the M2−M3 loop upward closer to the β1-β2 loop (Fig. 3d). Despite considerable conformational alterations in the ECD, the TMD of C1 state adopts a similar conformation as GLIC$_{pH5.5}$ structure, (Fig. 3c). Compared to C1, C2 state adopts a similar conformation, except that the upper part of M2 rotates counter-clockwise. Remarkably, the M2−M3 loop of the C2 state moves away from the pore axis with a maximum backbone displacement (~4.6 Å) observed at P250 (Fig. 3e). Likewise, the M2

helices in the O state undergo the largest counter-clockwise rotation (maximally 24° at T20´) with further outward movement of the M2−M3 loop (Fig. 3e and Supplementary Movie 2). The M2−M3 loop of GLIC$_{pH2.5}$-O adopts a conformation similar to GLIC$_{pH4.0}$-O (Fig. 3c). Thus, a systematic cascade of conformational changes is observed in both ECD and TMD of GLIC when the pH is reduced from neutral to acidic conditions.

## The ion permeation pathway and MD simulations of nanodisc-reconstituted GLIC reveal various intermediate non-conductive and open conformations

The TMD in all our structures exhibits the highest quality corresponding maps with a resolution better than 2.5 Å, as evident from the local resolution (Supplementary Figs. 2b, g, 5b, 7–9b, 13b). This facilitates the building of precise models at the TMD, particularly at the pore lined by M2 helices from five subunits (Supplementary Figs. 2e, j, 5e, 7–9e, 13e). In GLIC, residue I9´ (I233) serves as an activation gate, while the location of the desensitization gate remains unclear[60]. Multiple constrictions at the pore of GLIC$_{aso-pH7.5}$ and GLIC$_{pH7.5}$ are observed, with the most significant constrictions at I16´, I9´, and T2´. The highest constriction is observed at I9´ with a radius of ~2.24 Å (Fig. 4a, f and supplementary Fig. 14a, c). The pore profile of our apo state is distinct from the GLIC$_{detergent-pH7.5}$ (Supplementary Fig. 14c). In detergent-solubilized GLIC, the pore is more constricted at the aforementioned residues in the apo state, with an additional constriction at S6´ (Supplementary fig. 14c). The wider pore in our apo state is consistent with previous study[53]. However, the radius at I9´ falls below the radius of a hydrated Na⁺ ion (2.76 Å)[61], reflecting a non-conducting state (Fig. 4f and supplementary Fig. 14c). The subtle counter-clockwise twist at the upper part of M2 of GLIC$_{pH5.5}$ has minimal influence on the radius at I16´, I9´ and T2´, and shows an overlapping pore profile similar to that of GLIC$_{pH7.5}$ reflecting another non-conducting state (Fig. 4b, f). However, the pore of GLIC$_{pH4.0}$-C1 state is distinct from our pH 7.5 and 5.5 structures. The overall pore is wider, particularly at I16´, I9´, and T2´, with radii of 2.80, 2.39, and 2.67 Å, respectively (Fig. 4c, f). The pore below I9´, particularly at T2´, is wider than the existing putative open state (Supplementary Fig. 14c). Further widening of the pore is observed in GLIC$_{pH4.0}$-C2 structure with pore radii at I16´, I9´ and T2´ of 3.02, 2.55, and 2.85 Å, respectively (Fig. 4d, f). However, the narrower pore at I9´ in C1 and C2 states would restrict hydrated Na⁺ ions to pass through. In the GLIC$_{pH4.0}$-O state, a large twist at the upper part of M2 makes the pore wider with radii of 5 and 2.94 Å at I16´ and I9´, respectively (Fig. 4e, f). Notably, the radius of the pore at T2´ is 2.71 Å, which is smaller than the C2 state but marginally wider than the GLIC$_{detergent-pH4.0}$-O state (Fig. 4f and supplementary Fig. 14c). Similarly, the pore at S6´ of GLIC$_{pH4.0}$-O is wider than the existing GLIC$_{detergent-pH4.0}$-O state. Nevertheless, the widening at the pore makes the radius, particularly at I9´, larger than a hydrated Na⁺ ion (Fig. 4f). Using a previously modeled GLIC structure as a reference[8], a number of water molecules were incorporated in water-like densities of the corresponding high-resolution maps. Several layers of water molecules are observed in the pore of GLIC$_{pH4.0}$-O and GLIC$_{pH2.5}$-O structures, particularly in the upper and lower parts of I9´ (Fig. 4g and supplementary Figs. 14d, e, 15a). Four layers of loose water pentagons (pentagons 1–4) are observed above I9´ and extend to T20´ (Fig. 4g and supplementary Fig. 15a), while this region is occupied by detergent molecules in GLIC$_{detergent-pH4.0}$-O (Supplementary Fig. 15b). The water molecules in pentagons 2–4 are stabilized by polar interactions with water molecules present in the adjacent layer (Supplementary Fig. 15a). Water molecules in pentagon 1 do not involve in any polar interactions with each other. However, several water- or ion-like densities are observed in the vicinity of pentagon 1, which might stabilize the water (Supplementary Fig. 15a). Additional two layers of loose water pentagons (pentagons 5-6) are observed below I9´ (Fig. 4g and Supplementary Fig. 15a) which were also identified previously

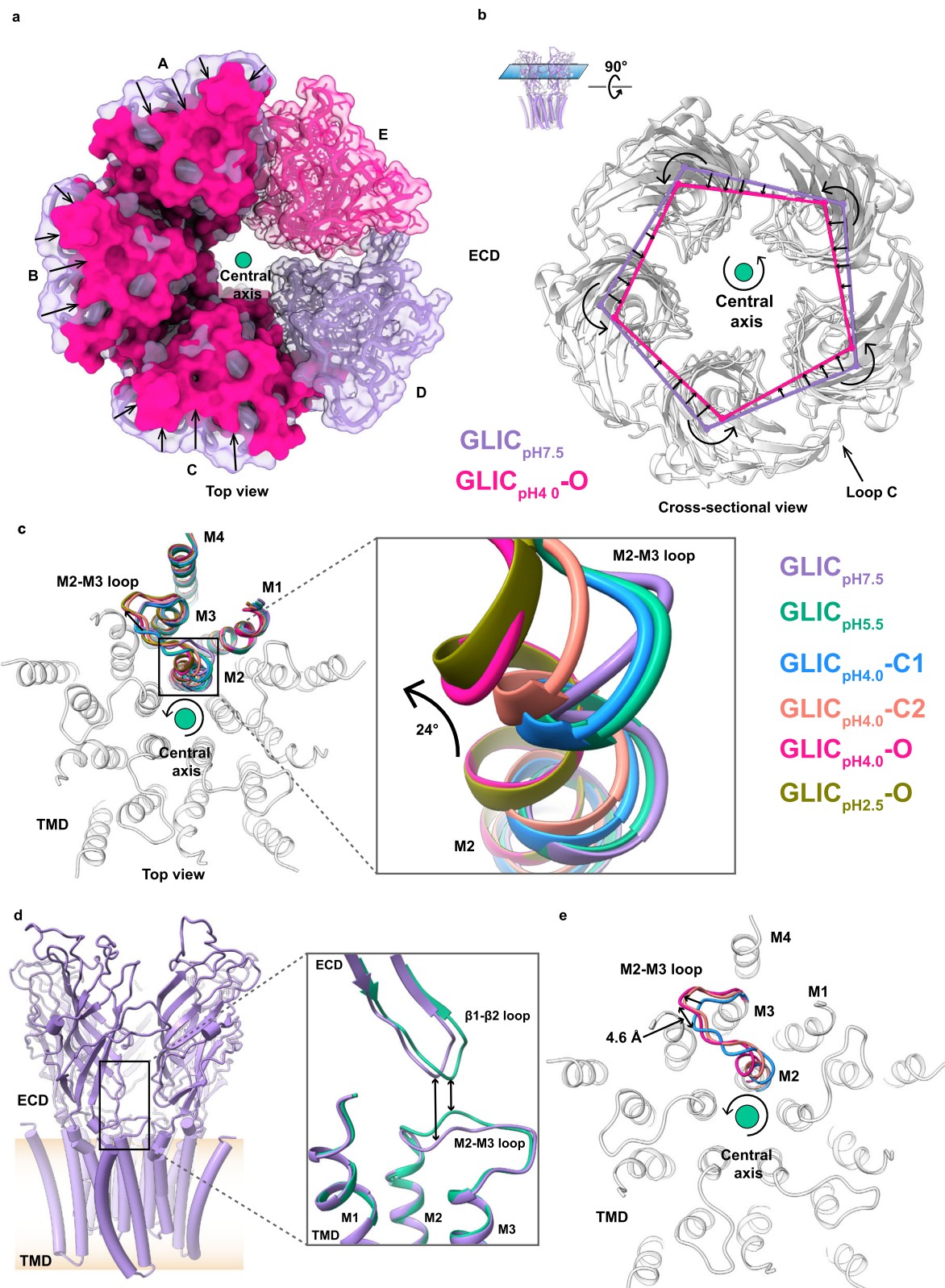

(Supplementary Fig. 15b). The density of the bottom-most water pentagon (pentagon 6) is also visible in the pore of GLIC$_{pH5.5}$ and GLIC$_{pH4.0}$-C1 maps (Supplementary Fig. 14f, g). These water molecules are stabilized by polar interactions with the side chains of S6´ and T2´. Few strong densities are present along the central axis of the pore of GLIC$_{pH4.0}$-O that could belong to cations (Supplementary Fig. 14h). However, because of their overlap with the symmetry axis, no ions

were modeled. Notably, the pore radii, hydration profiles, and the additional densities at the pore axis of GLIC$_{pH2.5}$-O are similar to GLIC$_{pH4.0}$-O.

To further define the functional states of our structures, MD simulations were performed on GLIC$_{pH7.5}$ and GLIC$_{pH4.0}$-O structures. Each structure was embedded within a bilayer consisting of the same ratio of DOPE, POPS, and POPC (2:1:1) used for cryo-EM. In both

**Fig. 3 | Comparison of GLIC in the apo and protonated states. a** Top view (from the extracellular side) of ECD shows the compaction of GLIC$_{pH4.0}$-O (deep pink) compared to GLIC$_{pH7.5}$ (purple). The subunits A–C of both structures are super-imposed and are shown in the surface representation. For clarity, the GLIC$_{pH7.5}$ surface is made transparent. Subunits D and E are shown only for GLIC$_{pH7.5}$ and GLIC$_{pH4.0}$-O, respectively, in the transparent surface, licorice, and stick representations. Arrows indicate the direction of compaction. **b** Cross-sectional view showing compaction and counter-clockwise rotation of ECD in GLIC$_{pH4.0}$-O compared with GLIC$_{pH7.5}$. The centroids of each subunit were calculated and connected using the axes between the centroids to show the overall compaction of ECD in GLIC$_{pH4.0}$-O. Axes and centroids are colored according to their respective structures. Arrows indicate the rotation and direction of the movements. The central axis is shown in green. **c** Top view showing the superposition of TMD in the apo and protonated states. Only the single subunit is colored as follows: GLIC$_{pH7.5}$ (purple),

GLIC$_{pH5.5}$ (green), GLIC$_{pH4.0}$-C1 (dodger blue), GLIC$_{pH4.0}$-C2 (salmon), and GLIC$_{pH4.0}$-O (deep pink), GLIC$_{pH2.5}$-O (olive). For clarity, other subunits are shown from a representative structure and colored gray. The inset shows a zoom-in view of the upper part of the M2 showing gradual counter-clockwise movement. Arrows indicate the direction of rotation in M2 and the outward movement of the M2–M3 loop. **d** Overall structure of GLIC (purple) showing β1-β2 loop and M2–M3 loop location (black rectangle). The inset shows the comparison of the β1-β2 loop and the M2–M3 loop in GLIC$_{pH7.5}$ (purple), GLIC$_{pH5.5}$ (green). Bidirectional arrows indicate approximate differences in the distance between these loops. The membrane bilayer is shown as a gradient of apricot color. **e** Top view of TMD showing the outward movement of the M2–M3 loop in pH 4 structures. The M2–M3 loops are colored according to their respective structures. The central axis is shown in green, and the arrow surrounding it shows the overall direction of rotation from C1 to C2 to O.

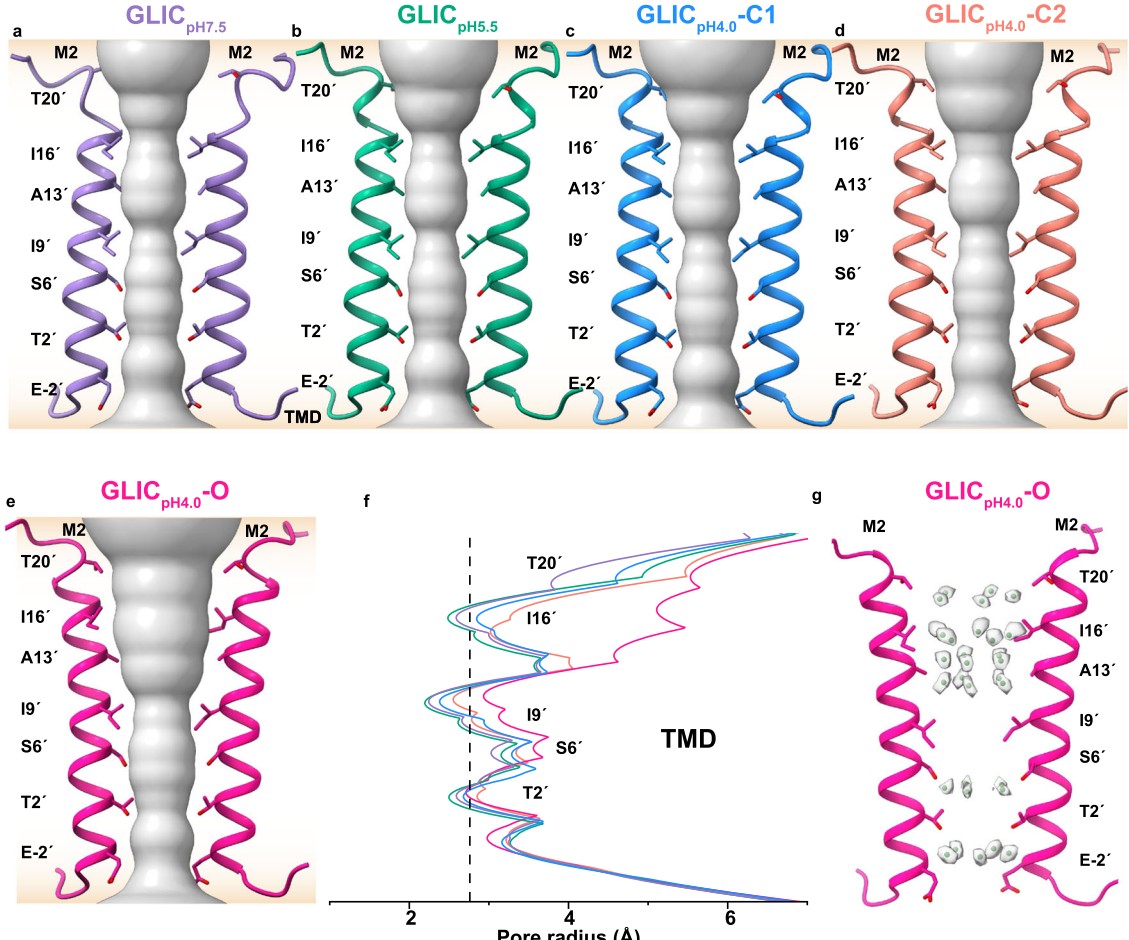

**Fig. 4 | The ion permeation pathway.** Pore profiles (surface representation in gray) were calculated using the HOLE program for (**a**) GLIC$_{pH7.5}$ (purple), (**b**) GLIC$_{pH5.5}$ (green), (**c**) GLIC$_{pH4.0}$-C1 (dodger blue), (**d**) GLIC$_{pH4.0}$-C2 (salmon), and (**e**) GLIC$_{pH4.0}$-O (deep pink). The membrane bilayer is shown as a gradient of apricot color. The M2 helices from the two diagonal subunits are shown in a cartoon for

clarity, and the pore-lining residues are shown as sticks. **f** Comparison of the pore radius plotted against the distance along the pore axis. Curves have the same color key as in (**a**–**e**). The vertical black dotted line represents the approximate radius of a hydrated Na$^+$ ion. **g** The cryo-EM densities (gray, transparent) of water molecules (green sphere) observed in GLIC$_{pH4.0}$-O are shown.

structures, the simulations predicted stable pore radii without any major changes during 200 ns equilibration (Fig. 5a, b). In GLIC$_{pH7.5}$, the pore at I9′ remains de-wetted during the entire simulation, which is supported by the presence of an energetic barrier for the water molecules at that region (Fig. 5c). However, the same extent of simulations of GLIC$_{pH4.0}$-O revealed no sign of an energetic barrier for water at I9′ (Fig. 5d). MD simulations in the presence of external potential show a similar de-wetted environment around I9′ throughout, reflected by no permeation events of Na$^+$ ions in GLIC$_{pH7.5}$ (Fig. 5e). Several

Na$^+$ ion permeation events were observed when the same experiment was performed on GLIC$_{pH4.0}$-O (Fig. 5f). Therefore, with the assistance of MD simulations, our cryo-EM structures have demonstrated that the GLIC$_{pH7.5}$ corresponds to a non-conductive state, whereas the GLIC$_{pH4.0}$-O represents an open conductive state. Additional MD simulations on C1 and C2 structures exhibited the de-wetted environment around I9′ during the entire simulation, which is supported by the presence of an energetic barrier for the water molecules, indicating non-conductive conformations (Supplementary Fig. 16).

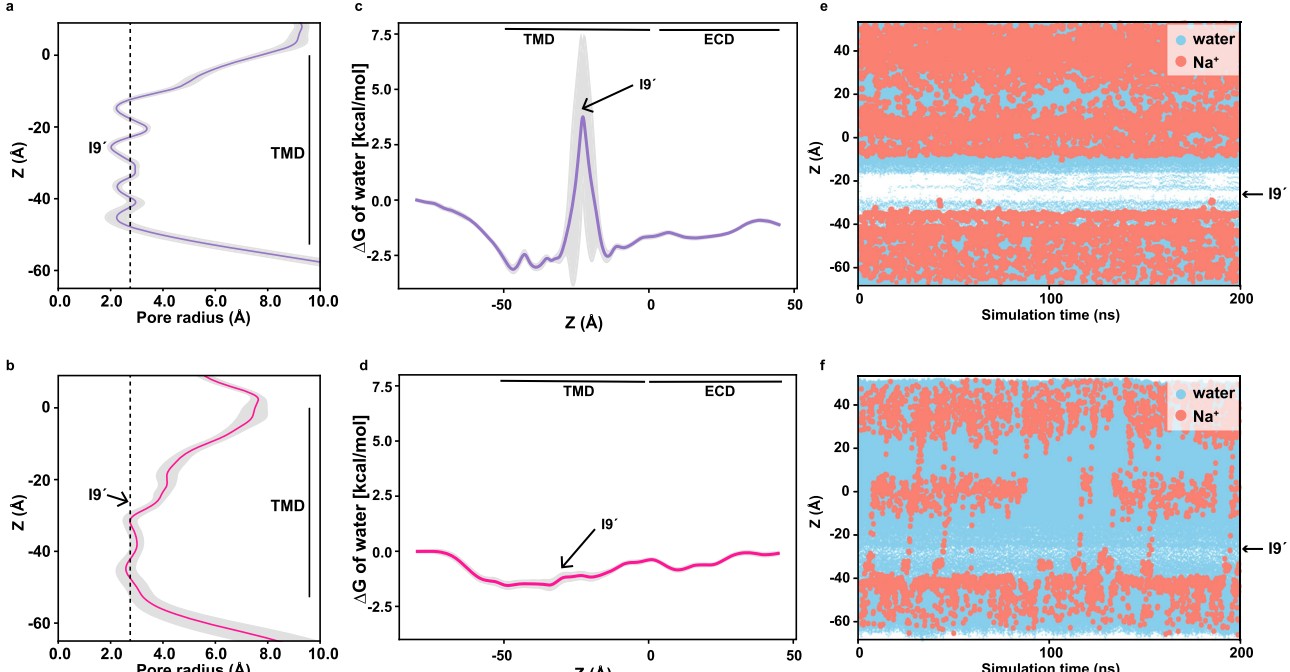

**Fig. 5 | Molecular dynamics simulations to investigate the pore hydration profile. a, b** The mean pore radius of GLIC_pH7.5 (purple) and GLIC_pH4.0-O (deep pink) along with the corresponding standard deviation in gray (simulation average and standard deviation from one out of three independent equilibrium simulation runs of 200 ns with a sampling interval of 0.1 ns). Black dashed lines represent the diameter of one water molecule. **c, d** Free-energy profiles of a water molecule along the central axis are shown for the corresponding structures with standard deviation in gray. **e, f** Time series of the water molecules (blue) and Na⁺ ions (salmon) are plotted for a simulation time of 200 ns for each corresponding structure. The location of I9′ is pointed out by an arrow. White stretches reflect regions lacking water.

## Complex protein–lipid interactions manifest across various states

A careful analysis of our maps showed several lipid densities at various locations adjacent to the TMD, under all pH conditions. Truncated phospholipids (PC in GLIC_aso-pH7.5 and PE in other structures) were placed in partially resolved non-protein lipid-like densities. Several lipid-like weak densities were either modeled as a single acyl chain or disregarded to prevent overfitting. The number of lipid molecules varies in each structure, reaching a maximum of nine lipids (Lipids 1–9) bound to each subunit. Lipids 1, 2, 6, 9 and 3–5, 7, and 8 are located at the upper and lower leaflets, respectively (Supplementary Figs. 17a–g, 18). Lipid 1 undergoes a subtle positional fluctuation in most of our structures (Supplementary Fig. 19a). In GLIC_aso-pH7.5, the head group of Lipid 1 is involved in polar interaction with the backbone amide group of M252 located at the M2–M3 loop and this interaction is not observed in GLIC_pH7.5 and GLIC_pH5.5 (Fig. 6a). However, in C1 state the head group of Lipid 1 interacts with T253 located at the M2–M3 loop and the substituent amido group wedges into the subunit interface and interacts with N200 (Fig. 6b). In C2 and O states, Lipid 1 density is disordered, allowing to build a single acyl chain in the map (Supplementary Fig. 17e, f). Lipid 2 is involved in polar interaction with the hydroxyl group of Y194 located at the pre-M1 in all structures (Fig. 6c). In pH 4.0 states, an acyl chain of Lipid 2 wedges into a cavity formed by M1, M3, and M4 helices (Fig. 6d). The head groups of Lipids 3 and 4 are positioned within a distance of 4 Å from R287 (Fig. 6e). Lipid 5 interacts with Y278 located at the M3 helix in all structures, whereas an additional interaction with an alternative conformation of R293 is observed only at acidic pH (Fig. 6f). Interestingly, the head group of Lipid 2 interacts with R118 located at the ECD of GLIC_pH7.5 and all pH 4.0 states, with the strongest interaction observed in state O (Fig. 6g, h). Lipids 6, 7, and 8 are observed at the new locations which are primarily stabilized by hydrophobic interactions. In C2 and O states, a new density for Lipid 9 is observed at the subunit interface in the vicinity of the pre-M1

helix (Fig. 6g, h and supplementary Fig. 17e, f). The ester oxygen in Lipid 9 interacts with hydroxyl oxygen of Y194 (Fig. 6g). In C2 state, the head group of Lipid 9 is oriented toward the pre-M1 and M2–M3 loops and involves in polar interactions with the backbone amide groups of Y194, F195, and P250 (Fig. 6g). Due to the subtle movement of the head group, Lipid 9 no longer interacts with P250 in O state (Fig. 6h). Compared to GLIC_pH4.0-O, very similar lipid locations and interaction networks are observed in our GLIC_pH2.5-O structure (Supplementary Figs. 17, 18g). Therefore, this study provides a detailed overview of complex protein–lipid interactions in GLIC.

## Discussion

Lipid molecules are found in several pLGICs and have implications in their functional role[28–36]. As a result, there is an increasing demand for methods to characterize membrane protein structures in physiologically relevant lipid environments[12,26,62]. However, detergents are often preferred to solubilize membrane-embedded proteins due to the difficulty of direct extraction from the lipid bilayer. However, detergents may strip away lipids that interact directly with the protein, thereby affecting its function[63]. In this study, we employed one of the most effective methods in investigating the structural conformations of GLIC in lipid environments, which involves solubilization of protein in detergent followed by reconstitution in lipid nanodiscs using scaffolding proteins. The size of the nanodisc scaffold affects the structure of pLGICs[64]. Structural analysis from previous studies suggested that the maximum diagonal distance of M4 is below 75 Å in GLIC, which was recently confirmed by HS-AFM[2]. Thus, in our current structural investigation, MSP1E3D1 is used as a scaffolding protein. MSP1E3D1 forms nanodiscs with a maximum diameter of 129 Å[65,66], suggesting that it will be appropriate for this study. A similar conformation of GLIC_pH7.5 and GLIC_aso-pH7.5 confirms the minimal impact of distinct lipid compositions on the GLIC structure in the apo state. A comparison between the detergent-solubilized cryo-EM and crystal structures of GLIC at pH 7.0,

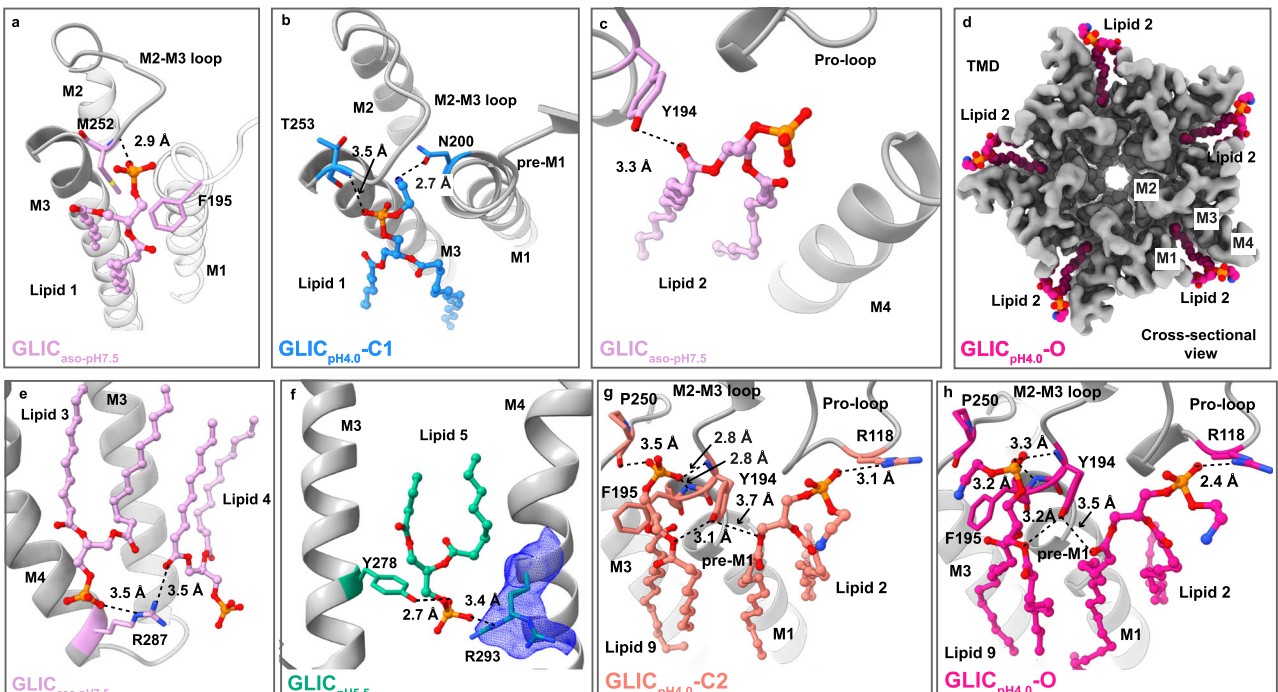

**Fig. 6 | GLIC-lipids interaction pattern.** Interactions of Lipid 1 with (**a**) GLIC$_{aso-pH7.5}$ (plum) and (**b**) GLIC$_{pH4.0}$-C1 (dodger blue), (**c**) Lipid 2 with GLIC$_{aso-pH7.5}$ are shown. Lipids are depicted in ball-stick representation. **d** Top view of the TMD (EM density, gray) shows wedging of Lipid 2 (ball-stick) inside the cavity formed by M1, M3, and M4 helices. The wedging of this lipid is observed in acidic pH conditions, and the lipids from GLIC$_{pH4.0}$-O (deep pink) are shown here as a representative. **e** The interactions of Lipids 3 and 4 with R287 in GLIC$_{aso-pH7.5}$ (plum) are shown. **f** The

interactions of Lipid 5 with Y278 located at M3 helix and with an alternative conformation of R293 on M4 in GLIC$_{pH5.5}$ (green) are shown. The cryo-EM density for R293 is displayed as blue mesh. Interactions of Lipids 2 and 9 in GLIC$_{pH4.0}$-C2 (salmon) (**g**) and in GLIC$_{pH4.0}$-O (deep pink) (**h**) are shown. Lipids are shown in ball and stick representations, whereas interacting residues are shown as sticks with corresponding colors. Protein is represented as a cartoon (gray).

reflects moderate conformational changes observed with backbone (Cα) ECD RMSD ≤1.4 Å[46], indicating the conformational similarities irrespective of the tools used for structure determination. However, our apo state exhibits significant conformational differences, particularly in the ECD, compared to detergent-solubilized GLIC, underscoring the pivotal role of lipid environments in influencing GLIC conformation. Compared to GLIC$_{pH7.5}$, the ECD in GLIC$_{pH5.5}$ undergoes a counter-clockwise rotation and compaction (the solvent accessible area is 48,666 Å²) with an increased buried surface area. Further analysis of the ion permeation pathway shows a wider pore in both of our pH 7.5 and pH 5.5 structures compared to the existing apo state in detergent, albeit narrower than a hydrated Na⁺ ion and non-conductive, as confirmed by MD simulations (Fig. 4f and supplementary Figs. 14c, 5e). Notably, clockwise rotation and expansion of ECD were observed in previously determined high-resolution structures of pLGICs in a resting-like state compared to agonized states[14,21,51]. Therefore, our GLIC$_{pH7.5}$ and GLIC$_{aso-pH7.5}$ represent a state closer to the resting conformation. Likewise, based on the compaction, counter-clockwise rotation in ECD, and closed pore, the GLIC$_{detergent-pH7.5}$ and our GLIC$_{pH5.5}$ structures have likely been captured at intermediate non-conductive states. The intermediate conformation of our GLIC$_{pH5.5}$ is also supported by the inward movement of Loop F toward the central axis and reduced distance between β1-β2 loops and the M2–M3 loops, which was also observed in the putative open states[7,8].

Reduction of pH to 4.0 stabilizes the C1, C2, and O states with population distributions on cryo-EM grid of 50, 15, and 35%, respectively. These states adopt unique conformations with characteristic conformational differences observed in various regions. The ECDs in these structures show a higher extent of counter-clockwise rotation compared to the pH 5.5 structure, and the pores appear to be wider (Fig. 4f). Compared to the C1 state, the ECD shows subtle overall conformational changes in C2. However, unique changes are observed in

the TMD, where the upper portion of the M2 undergoes counter-clockwise rotation leading to an approximately 4.6 Å outward movement of the M2–M3 loop from the pore axis, which is the characteristic mechanism of the pLGICs activation[7,28,67–69]. Despite the conformational changes in both C1 and C2 states, the pore remains constricted and dewetted, confirmed by MD simulations. The O state adopts a unique conformation among all the aforementioned structures. Further counter-clockwise rotation in the ECD and outward movement of the M2–M3 loop are observed in the O state compared to C2 (Fig. 3e and Supplementary Movie 3). Compared to the apo state, a significant counter-clockwise rotation is observed in the M2 helices of the O state, particularly in the upper part of the pore (Fig. 3c), accompanied by 8° outward tilt from the pore axis (Supplementary Fig. 14i). The calculated radius of the pore at the gate (I9´) is larger than a hydrated Na⁺ ion. However, below I9´, particularly at T2´, the pore becomes narrower with a radius of 2.71 Å compared to the C2 state (Fig. 4f). Although this region is narrower than a hydrated Na⁺ ion, due to the polar nature of T2´, the ions could be present in a partially dehydrated form[10]. A small fluctuation of the sidechain location of T2´ away from the pore axis could allow the ion to pass through. Furthermore, careful analysis revealed several water molecules at the pore of our O state, which are either absent or reduced significantly in other structures. The absence of previously observed detergent molecules in the pore of our open structure increases the occupancy of water molecules above the I9´, which was also predicted previously by simulations[8]. These water pentagons are loosely packed and might be stabilized by coordination with ions that were not built in our structures (Supplementary Fig. 14h). Below I9´, two layers of tighter water pentagons are present in our O state structure near S6´ and T2´, which has been reported previously[8], reflecting elevated hydration at the pore. This is validated by our MD simulations experiments, which show an increase in the number of water molecules without any energetic barrier (Fig. 5d). Further MD

simulations revealed several $Na^+$ ions permeation events indicating the conductive conformation of our O state (Fig. 5f). The overall conformation of our open state is close to the GLIC$_{detergent-pH4.0}$-O, with backbone (Cα) RMSD of ECD being 1.1 Å. However, compared to our open state, the ECD of GLIC$_{detergent-pH4.0}$-O, moves toward TMD to a higher extent in addition to further counter-clockwise rotation at the M2 (Supplementary Fig. 19c, d). Nevertheless, the M2-M3 loop adopts a very similar conformation in both structures (Supplementary Fig. 19d). The pore in GLIC$_{detergent-pH4.0}$-O is wider with a radius of 3.2 Å at I9′ (Supplementary Fig. 14c), which is possibly due to the trapped detergent molecules. MD simulations of putative GLIC structure without detergents showed various extents of constrictions while the pore remained hydrated[3,7,60]. Hence, GLIC$_{detergent-pH4.0}$-O might have been captured in another form of open states. Intriguingly, only 35% of the population of GLIC adopted an open state at pH 4.0 in this study, while the rest likely represent pre-open and other intermediate states. Prolonged exposure to an activating condition should stabilize GLIC in the desensitized state, where the ECD should adopt the activated conformation, while the pore remains non-conductive. This prompts consideration as to whether the C1 and/or C2 states represent desensitized states rather than pre-open states. It is evident from previous functional studies that the open probability and rate of desensitization increase with stronger activating conditions, such as reducing pH below 4.0[39,70,71]. The structure of GLIC at pH 2.5, lower than the pH$_{50}$ of liposome-reconstituted GLIC, demonstrates that all intact particles on the cryo-EM grid adopt an open state similar to the O state (Supplementary Fig. 12). This observation implies that both C1 and C2 precede the O state. Moreover, according to the current understanding of GLIC and other eukaryotic pLGICs, the desensitized state has constricted pore below 9′, particularly near the −2′ region[3,27,72], which is absent in both C1 and C2 states. MD simulations of C1 and C2 states show dehydration at I9′ and A13′ in both structures, which is similar to our apo state. Therefore, we propose that C1 and C2 likely represent intermediate states that precede the open state.

While GLIC exhibits less sensitivity to the lipid composition for the agonist-induced cation flux[63,73], additional experimental evidence is necessary to comprehend the impact of state-dependent lipid–protein interactions. In the recent cryo-EM structure of detergent-solubilized GLIC, two lipids in the upper leaflet and three lipids in the lower leaflet were observed in each subunit, which were co-purified during sample preparation[47]. These lipids were observed upon re-processing the data from their earlier study, where no lipids were resolved, and the structures represented non-activating conformations[46]. Notably, a small subset of data yielded a high-resolution closed state with visible lipid densities, suggesting the influence of low lipid occupancy on stabilizing non-activating conformational states in their previous study[46]. In our structures, numerous lipid molecules are identified near the TMD, with a higher number of lipids present at the lower leaflet (Supplementary Figs. 17, 18). The positions of Lipids 2, 3, and 4 closely resemble those in existing crystal structures[7,8] (Supplementary Fig. 17h). Lipids 1–5 were also observed in a recent cryo-EM structure of GLIC[47], with marginal differences in the locations of Lipids 1 and 5 (Supplementary Fig. 17j, k). Interestingly, Lipid 5, located at the lower leaflet, interacts with an alternate conformation of R293 in our structures, a feature not observed previously (Fig. 6f). Mutation of this residue causes statistically significant changes in the pH$_{50}$ of GLIC, indicating the importance of this protein–lipid interaction in the channel activation[63]. A similar effect on pH$_{50}$ was noted with the mutation of R287[63], which directly interacts with other lipids located at the lower leaflet in our structures. Furthermore, Y278, which interacts with Lipid 5 in all of our structures, was predicted to be involved in the interaction with neurosteroids which potentiate the GLIC activation[74]. Although the lipids in the lower leaflet interact with the important residues of GLIC, the locations and conformations of those lipids are highly conserved, showing no clear evidence of state-dependent regulatory roles, consistent with previous study[47].

In contrast, the lipids in the upper leaflet have more dynamic locations and interaction networks. The polar interactions of the Lipid 2 head group with the neighboring residues are conserved[8] and speculated to be responsible for stabilizing the open form of GLIC[75]. Interestingly, our current study showed an additional density for one of the acyl chains in all pH 4.0 states, which transverses to the cavity formed by residues present in the upper parts of M1, M3, and M4 (Fig. 6d). Remarkably, general anesthetics, such as propofol and desflurane, were found at the same cavity and stabilized the open state of GLIC (Supplementary Fig. 19e)[42,50]. Therefore, the penetration of the acyl chain of Lipid 2 at the activating conditions (pH 4.0 or lower) indicates its potential state dependence and role in channel opening and stabilization. Lipids 1 and 9 exhibit the most dynamic conformations in our structures. Upon lowering pH, the head group of Lipid 1 progressively moves into an intersubunit cavity, formed due to rotameric change in the M252 sidechain, leading to interactions with N200 and T253 located in the pre-M1 helix and M2-M3 loop, respectively (Fig. 6b). Interestingly, the outward movement of M2-M3 loop and downward movement of M252 sidechain in C2 and O states destabilizes Lipid 1 (Supplementary Fig. 19f). The absence of Lipid 1-N200 interaction and outward movement of M2 helix facilitate the interaction between N200 and E243, which is only observed in our open state (Supplementary Fig. 19g). E243 in GLIC and the equivalent position of N200 in the glycine receptor play a critical role in stabilizing the open state, as confirmed by the mutational studies[37,38,75,76]. In addition, the intersubunit interaction between the equivalent N200-E243 in the glycine receptor was shown to be important for channel gating[77]. While our study reveals that the conformation of Lipid 1, particularly the head group, is state-dependent, its role in GLIC channel opening is unclear and requires further investigation. Recently, MD simulations predicted the presence of an additional lipid in the open state of GLIC at the upper leaflet[47], which lies between two subunits, and the predicted location is close to our Lipid 9 (Supplementary Fig. 17l). The sidechain of F195, located at the pre-M1, rotates 17.7° compared to our C1 state to accommodate the acyl chain of Lipid 9, which is involved in interaction with the backbone amide group of F195 (Supplementary Fig. 19h and Fig. 6g, h). A similar hypothesis has been drawn previously after performing MD simulations on the putative open state of GLIC[47]. Mutation of F195 to alanine led to loss of function, suggesting the importance of this residue in gating[9]. We observed a polar interaction between Lipid 9 and P250 located at the M2-M3 loop of GLIC$_{pH4.0}$-C2, which has been predicted previously[47] (Fig. 6g). Interestingly, the proline residue located in a similar location in M2-M3 loop of another pLGIC has been implicated in channel gating[78]. However, due to the subtle positional alteration of the head group, Lipid 9 does not interact with P250 in our open state but maintains polar interactions with Y194 and F195 located in pre-M1 (Fig. 6h). Notably, in all the structures, Y194 consistently interacts with Lipid 2, which in turn interacts with R118 situated at the Pro-loop of ECD. Therefore, Lipid 9 may play an important role in gating by bridging the M2-M3 loop, Pre-M1, and other parts of the ECD at the ECD-TMD interface. In ELIC, a lipid molecule which triggers the channel opening[13] is observed in a location similar to our Lipid 1, with its head group positioned between Lipid 1 and Lipid 9 (Supplementary Fig. 19b). Similarly, in GluCl and glycine receptors, the antiparasitic drug ivermectin is located at a position similar to that of Lipid 9 and stabilizes an open or desensitized state[20,27,79,80]. Therefore, our data reveals that Lipid 9 is another important lipid, which binds to the GLIC in a state-dependent manner. As mentioned earlier in the O state, a critical triad is formed by bridging D32-R192-D122 located in β1-β2, pre-M1, and Pro-loop. In our O state, another long-range bridge is formed between Pro-loop and pre-M1 via Lipids 2 and 9, which might be critical for gating. Therefore, our study provides insights into the conformation-specific lipid–protein

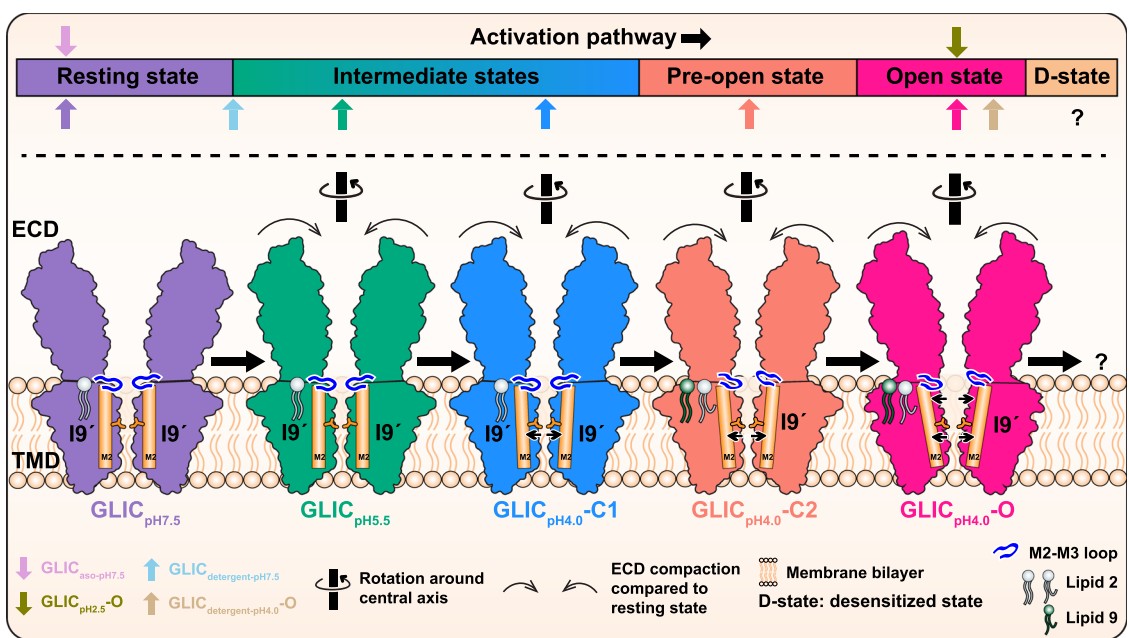

**Fig. 7 | Schematic of the gating mechanism of GLIC.** The activation pathway of GLIC is shown with a resting state represented by our structures at pH 7.5, two putative intermediate states observed at pH 5.5 and pH 4.0 (C1), a putative pre-open state with a wider pore at I9′ and outward movement of M2−M3 loop (C2), and an open state represented by the O structure at pH 4.0 with a maximally open pore. Lipid 2 is observed in all the structures. An acyl chain of Lipid 2 wedging into a cavity is observed only in C2 and O states. Lipid 9 is observed in both putative pre-open (C2) and open states. In the open state, the pore is wider than all other structures, along with a counter-clockwise rotation of the upper part of the M2 helix and outward motion of the M2−M3 loop. All protonated structures show counter-clockwise rotation and compaction of the ECD compared to the resting state. The representative closed (PDB: 4NPQ) and putative open (PDB: 4HFI) structures solved by crystallography are placed on the activation pathway along with all the structures in the current study. A desensitized state (?) is not observed in our study. A key for various elements used for depiction is given at the bottom.

interactions, particularly at the upper leaflet, which may play a crucial role in stabilizing the various functional states.

Due to the complexity of the gating mechanism, which involves several unresolved intermediate states[53], the assignment of functional states in a sequence along the activation pathway is challenging. Our apo structures, characterized by an expanded and clockwise twisted ECD compared to GLIC_detergent-pH7.5, represent a state closer to the resting state (Fig. 7), whereas GLIC_detergent-pH7.5 represents a state along the activation pathway. A structural comparison of GLIC_pH5.5 and GLIC_detergent-pH7.5 shows that the backbone (Cα) RMSD of the ECD is 1.2 Å, reflecting the similarities between these two conformations. However, due to the inward movement of loop F and upward movement of the M2−M3 loop, it likely represents a state that falls ahead of the GLIC_detergent-pH7.5 along the activation pathway (Fig. 7). This is followed by the C1 state, which also exhibits counter-clockwise rotated ECD and closed pore. The overall conformation of C2 state is similar to C1, particularly at the ECD. However, the M2−M3 loop moves away from the central axis and adopts a similar conformation as observed in a previously solved locally closed 2 (LC2) pre-open state of GLIC[37,75] (Supplementary Fig. 19i). Recently, fluorescent quenching experiments revealed similar compaction of ECD and outward movement of the M2−M3 loop upon protonation while the channel was closed, which was referred to as the pre-active state[1]. Therefore, based on the previous study, our C2 state likely represents a pre-open state, which chronologically follows our C1 state. The O state at pH 4.0 and pH 2.5 represent the extreme end state of the activation pathway in our study and represent an open state. The GLIC_detergent-pH4.0-O state would be positioned after our open state in the activation flow chart, considering no effect of detergent molecules on the pore radius (Fig. 7).

In summary, we present cryo-EM structures of nanodisc-reconstituted GLIC at various pH conditions, which are distinct from the existing structures. Our structures provide a better understanding of proton-induced conformational transitions, the gating mechanism,

and state-dependent protein−lipid interactions in GLIC at the molecular level. Several lipids with high occupancies were assigned, showing differential interaction patterns in various states. Therefore, our investigation of GLIC in the lipid environment enriches our knowledge of the various intermediate states, including a putative pre-open state along the activation pathway and the complex cross-talk between protein and lipids.

The primary limitation lies in our inability to capture a desensitized state, despite the protein being at low pH conditions for an extended period. Cholesterol and DHA have been shown to increase the desensitization probability of GLIC[10,39]. Therefore, future studies could employ desensitization-enhancing modulators, such as cholesterol and DHA, in nanodisc preparation. Additionally, different scaffold proteins could be tested during the nanodisc preparation to explore their impact on the structural conformation, as indicated by a recent study highlighting the influence of scaffolding proteins on a pLGIC conformation[64]. While a previous study showed that the TMD does not possess the principal component of proton activation[38], the inhibitory effect of long-term exposure to internal low pH (pH_i) has been demonstrated[49,81], which potentially contributes to the observed slow transition rate along the activation pathway. Determination of GLIC structures in a more native-like environment, such as liposomes, might provide further insights into the gating mechanism in a near-native environment. However, upon activation, GLIC could also permeate protons and decrease the internal pH of liposomes, as speculated previously[81]. Recently, fumarate has been identified as the best positive allosteric modulator of GLIC[81], which could be used in the future to explore the possibility of stabilizing the desensitized state of GLIC.

## Methods
### Cloning, expression, and purification of GLIC
The GLIC-MBP gene was cloned into the pET26 vector using a protocol described previously[11]. Briefly, we transformed the construct into the *E.*

*coli* C41 cells. The successfully transformed cells were grown in Terrific Broth (TB) containing 50 µg/mL kanamycin at 37 °C until $OD_{600}$ reached 0.8, and cells were induced with 200 µM IPTG and further incubated at 18 °C. The cell pellet was harvested after 14–16 h and resuspended in ice-cold lysis buffer (20 mM HEPES, 150 mM NaCl, pH 7.5, supplemented with 1 mM Phenylmethylsulfonyl fluoride (PMSF), 1 µM Leupeptin, and 1 µM Pepstatin). The resuspended cells were lysed using Avestin Emulsiflex C3 homogenizer, and the membrane was isolated by centrifugation at 168,000 × g for 45 min. The membrane was then resuspended in a buffer containing 20 mM HEPES, 150 mM NaCl, pH 7.5, 10% glycerol, and flash frozen in liquid nitrogen upon uniform resuspension. The frozen samples were kept at −80 °C until further use.

The resuspended membrane was solubilized in 1% n-dodecyl-β-ᴅ-maltoside (DDM). The undissolved debris was removed by ultracentrifugation at 168,000 × g for 15 min. The supernatant was bound to 2 mL amylose resin (NEB) at 4 °C for 1.5–2 h. Subsequently, the mixture was loaded into a gravity flow column. The resin was washed with 40 mL wash buffer (20 mM HEPES, 150 mM NaCl, 1 mM EDTA, 0.05% DDM, pH 7.5) and protein was eluted in 10 mL elution buffer (20 mM HEPES pH 7.5, 150 mM NaCl, 20 mM Maltose, 0.05% DDM). Eluted protein was concentrated to 7–8 mg/mL using Amicon Ultra – 4 centrifugal filter units with 50 kDa cutoff. The MBP tag was removed using HRV-3C protease. The digested GLIC, MBP, and protease were separated using a superdex-200 increase column (GE Healthcare) pre-equilibrated with gel filtration buffer (20 mM HEPES pH 7.5, 150 mM NaCl, 0.025% DDM) in size exclusion chromatography (SEC) system from Cytiva ÄKTA purifier.

### Cloning, expression, and purification of MSP1E3D1

The MSP1E3D1 cloned in the pET28a vector was purchased from Addgene (Addgene plasmid #20066)[82]. The plasmid was transformed into *E. coli* BL21 cells and grown in TB with 25 µg/mL kanamycin followed by induction using 1 mM IPTG when $OD_{600}$ reached 0.8. Cells were grown at 37 °C and harvested after 6 h. The cells were resuspended in lysis buffer containing 20 mM HEPES, 150 mM NaCl, pH 7.5, 1 mM PMSF, and 10% glycerol, followed by homogenization using Avestin Emulsiflex C3 homogenizer. Debris was removed by ultracentrifugation at 168,000 × g for 45 min. The supernatant was bound to 2 mL Ni-NTA resin for 1 h and subsequently loaded into a gravity flow column. Bound beads were washed with 40 mL of buffer A (50 mM Tris, 300 mM NaCl, 1% Triton X-100, pH 8.0), buffer B (buffer A with 50 mM sodium cholate, pH 8.0), buffer C (50 mM Tris, 300 mM NaCl, pH 8.0), buffer D (buffer C with 30 mM Imidazole, pH 8.0), and buffer E (buffer C with 2 mM EDTA and 2 mM DTT, pH 8.0). Protein was eluted with 10 mL of the elution buffer (buffer C with 300 mM Imidazole, pH 8.0). Imidazole was removed by passing eluted protein through a PD-10 column equilibrated with 50 mM Tris, 100 mM NaCl, 0.5 mM EDTA, and pH 8.0. The purified MSP1E3D1 was concentrated at 10 mg/mL and used for the experiments.

### Nanodisc reconstitution and purification

Asolectin (10 mg/mL in chloroform from Avanti) or a lipid mixture of DOPE:DOPS:DOPC (2:1:1 by volume; all lipids in 10 mg/mL chloroform from Avanti) was dried in a stream of nitrogen and resuspended in low-salt buffer (20 mM HEPES pH 7.5, 150 mM NaCl). Freshly purified GLIC was used for nanodisc reconstitution in the ratio of 1:3:360:1800 (GLIC:MSP1E3D1:lipid:DDM) by mole was used for all sample preparations. The mixture was incubated at 4 °C for 1 h, and a minimal amount of Bio-Beads SM-2 adsorbents (Bio-Rad Laboratories Cat No: 1523920) were added to remove detergent. After overnight incubation with Bio-beads, the nanodisc-reconstituted GLIC was purified using gel filtration, as described previously, in the running buffers (20 mM HEPES pH 7.5, 150 mM NaCl, 1 mM EDTA; 10 mM MES pH 5.5, 150 mM NaCl, 1 mM EDTA; 10 mM sodium citrate pH 4.0, 150 mM NaCl, 1 mM EDTA, respectively).

### Cryo-EM grid preparation and data collection

The SEC fractions containing GLIC in the asolectin nanodisc at pH 7.5 were pulled together and concentrated to 1.4 mg/mL, while the SEC fractions for GLIC nanodiscs in DOPE:POPC:POPS were pulled together and directly used for grid preparation at concentrations of 0.2–0.5 mg/mL. GLIC nanodiscs sample at pH 2.5 was prepared by buffer exchange of pH 4.0 sample with 10 mM sodium citrate pH 2.5, 150 mM NaCl, and 1 mM EDTA. For the $GLIC_{aso-pH7.5}$ sample, quantifoil R1.2/1.3 Au 300 mesh grids were used, while in all other conditions, quatifoil R1.2/1.3 300 mesh Cu grids with a graphene monolayer were used. A volume of 3.5 µL of sample was applied to the EM grids at FEI Vitrobot Mark IV. All the grids were blotted once, except for $GLIC_{aso-pH7.5}$, which was blotted twice at 100% humidity and temperature of 4 °C and rapidly plunge-frozen into liquid ethane.

All data were collected using a FEI Titan Krios 300 kV Cryo-TEM with Gatan K2, K3, and Falcon4i direct electron detector equipped with an energy filter and slit width of 10 or 20 eV (Supplementary Table 1). For $GLIC_{aso-pH7.5}$, 5188 movies were collected, each with 40 frames per movie at 130,000× magnification in super-resolution mode using K2 detector. The physical pixel size was 1.064 Å/pixel, and the total dose was 40 e⁻/Å². For $GLIC_{pH5.5}$, 6625 movies were collected at a physical pixel size of 0.8452 Å/pixel with a total dose of 57 e⁻/Å², and at 165,000× magnification in super-resolution mode using the K2 detector. For $GLIC_{pH7.5}$ and $GLIC_{pH4.0}$, 14,001 and 14,076 movies were collected, respectively, at a physical pixel size of 0.76 Å/pixel with a total dose of 71 e⁻/Å², and at 165,000× magnification using Falcon4i detector. For $GLIC_{pH2.5}$-O, 6449 movies were collected at a physical pixel size of 0.8584 Å/pixel with a total dose of 74 e⁻/Å², and at 165,000× magnification using the K3 detector. The defocus range used in all datasets was −0.6 to −2.0 µm (Supplementary Table 1).

### Image processing and 3D reconstruction

MotionCor 2 (v1.6.3) was used for beam-induced motion correction in all datasets[83], and Ctffind4 (v4.1.13)[84] was used to estimate the Contrast Transfer Function (CTF). For all datasets, particles were picked either by blob picking, template picking, or using TOPAZ (v0.2.5a)[85]. Reference-free 2D classification was used to clean the picked particles, followed by ab initio 3D reconstruction and hetero-refinement to further clean and refine the map. For the $GLIC_{aso-pH7.5}$ dataset, a total of 81,808 particles were refined in CryoSPARC (v4.3.1)[86]. Subsequently, particles were transferred into RELION (v4.0.1)[58], where CTF refinement, 3D classification without alignment, 3D refinement, and Bayesian polishing were performed, which yielded a final map with 59,086 particles. Finally, post-processing and mask application generated a 3D sharpened map with a nominal resolution of 3.4 Å (Supplementary Fig. 1a). For the $GLIC_{pH7.5}$ dataset, a total of 7,888,849 particles were refined in CryoSPARC, followed by particle subtraction to remove signal of nanodisc. Subtracted particles were refined in cryoSPARC to improve the alignment. Aligned particles were transferred into RELION-4.0 and subjected to 2D classification, CTF refinement, 3D classification without alignment, Bayesian polishing, and 3D refinement. Finally, post-processing and mask application yielded a map with a nominal resolution of 2.9 Å from 152,938 particles (Supplementary Fig. 1b). For the $GLIC_{pH5.5}$ dataset, a total of 365,137 particles were refined in CryoSPARC. This dataset showed preferred orientations in the particle distribution, as evident from the angular distribution (Supplementary Fig. 4). To further refine the particles, Bayesian polishing was performed for side view particles, whereas top view particles were subjected to 2D classification (T = 4). Furthermore, several rounds of 3D classification without alignment (T = 4, 8, 32), particle subtraction, and non-uniform refinement were performed. Finally, post-processing and mask application generated a 3D map with

a nominal resolution of 2.7 Å using 61,526 particles. For the GLIC$_{pH4.0}$ dataset, 1,071,249 particles were refined using CryoSPARC. Subsequently, the particles were transferred into RELION, where 3D classification without alignment resulted in three unique classes. Each class was processed independently, where CTF refinement, Bayesian polishing, and 3D classification without alignment were performed. Finally, post-processing and mask application produced three unique 3D reconstructions with distinct conformations named C1, C2, and O state with nominal resolutions of 2.7, 3.3, and 2.7 Å, respectively. The population distribution of C1, C,2, and O state is observed to be 50, 15, and 35%, respectively. (Supplementary Fig. 6). For the GLIC$_{pH2.5}$-O dataset, a total number of 329,848 particles were refined in CryoS-PARC. Subsequently, particles were transferred into RELION, where CTF refinement, 3D classification without alignment (T = 4, 8), and Bayesian polishing were performed. Finally, post-processing and mask application generated a 3D map with a resolution of 2.6 Å using 98,776 particles (Supplementary Fig. 12). The local resolution map was calculated using ResMap (v1.1.4)[87]. Surface coloring of the density map was performed using the UCSF ChimeraX[88]. Both the gold standard Fourier shell correlation (FSC) curves and the model Vs map validation FSC curves for all models were plotted in OriginPro (v9.9.0.225), using the output of post-process jobs of RELION and the correlations calculated from EMAN (v2.99.47)[89].

## Model building, structure refinement, and figure preparation

The crystal structure of GLIC$_{detergent-pH7.5}$ (PDB ID: 4NPQ) was used as an initial model for GLIC$_{aso-pH7.5}$ model building (Supplementary Table 1). The initial model was aligned to our density map in the UCSF chimeraX. The aligned model was used for manual model building in the Coot program (v0.9.8.7)[90]. The manually built model was refined against the corresponding density map using the phenix.real_space_refine tool from the PHENIX package (v1.21rc1-4985)[91], with all default settings except full NCS constraints, fully refined NCS operators, and "max reasonable bond distance" set to 500. The final refined GLIC$_{aso-pH7.5}$ model was used as an initial model for all the remaining cryo-EM data and refined similarly. Molprobity web server (v4.5.2)[92] was used to analyze the stereochemistry of the final models. The protein surface area and interfaces were investigated using the PISA program integrated into the CCP4i2 software package (v1.0.2)[93]. The pore profile was calculated using the HOLE program (v2.2.005)[94]. The calculation of rotation angles involved initially aligning models through TMD. Subsequently, the angle formed by the two atoms of interest, using the horizontally coplanar point on the central axis as the apex, was determined by utilizing only the x and y coordinates from the PDB files. All the figures and movies were prepared in UCSF Chimera (v1.17)[95], ChimeraX (v1.7)[88], and Adobe Illustrator (v28.1).

## Molecular dynamic simulations

Cryo-EM structures of the GLIC in each state were embedded within bilayer membranes consisting of DOPE, POPS, and POPC with a 2:1:1 ratio with the CHARMM-GUI Membrane Builder in 11.6 × 11.6 × 15.8 nm³ simulation cells[96,97]. Similar to previous studies, several acidic residues such as E26, E35, E67, E75, E82, D86, D88, E177, and E243 were protonated, and H277 was doubly protonated to mimic the low pH conditions in case of GLIC$_{pH4.0}$-O[42,47,60]. Simulations were performed with GROMACS 2021[98]. The integration time-step was 2 fs. Bonds were constrained through the LINCS algorithm[99]. A verlet cut-off scheme was applied, and long-range electrostatic interactions were calculated using Particle Mesh Ewald method[100]. Temperature and pressure were maintained at 310 K and 1 bar during simulations, using the velocity-rescaling thermostat[101] in combination with a semi-isotropic Parrinello and Rahman barostat[102], with coupling constants of 1 and 5 ps, respectively. To best preserve the conformational state of each cryo-EM structure during simulations, harmonic restraints at a force constant of 1000 kJ mol$^{-1}$ nm$^{-2}$ were placed on protein backbone atoms,

and MD simulations ran for 50 ns. In the case of GLIC$_{pH4.0}$-O, the initial equilibration was extended using a previously developed simulation protocol that was found to aid in the relaxation of stable open states in pentameric LGICs[103]. The ion channel pore was kept open via flat-bottom harmonic cross-distance position restraints between selected Cα atoms of pore-lining residues (Thr at 20′, Ile at 16′, Ala at 13′, Ile at 9′, Ser at 6′, Thr at 2′, and Glu at −2′). The flat-bottom potential only acts when the Cα atoms of non-adjacent pore-lining residues come closer to each other than their distance in the open-state cryo-EM structure with a force proportional to the difference in distance (5000 kJ mol$^{-1}$ nm$^{-1}$). This setup allows the pore-lining side chains to relax. This additional restrained simulation step with pore restraints was performed for 200 ns. The final frame was used as the starting configuration for three independent repeats, initiated with different velocities. Pore water-free-energy profiles were computed for alternative conformations of the protein using the Channel Annotation Package[104], in each case based on 200 ns equilibrium simulations at physiological salt (150 mM NaCl) concentration. Simulation trajectories were analyzed at 100 ps intervals, with a bandwidth of 0.14 nm applied for water density estimation. Ion conduction was measured in 200 ns simulations for each backbone-restrained receptor structure at 150 mM NaCl concentration and in the presence of a +500 and −500 mV transmembrane potential difference, with positive potential on the cytoplasmic side and on the extracellular side, respectively. This was applied by imposing an external, uniform electric field in the membrane's normal direction. Simulation boxes, gromacs input files, and final frames of simulations can be downloaded from Zenodo https://doi.org/10.5281/zenodo.10792701.

## Reporting summary

Further information on research design is available in the Nature Portfolio Reporting Summary linked to this article.

## Data availability

Accession Numbers-The coordinates of the GLIC$_{aso-pH7.5}$, GLIC$_{pH7.5}$, GLIC$_{pH5.5}$, GLIC$_{pH4.0}$-C1, GLIC$_{pH4.0}$-C2, GLIC$_{pH4.0}$-O, and GLIC$_{pH2.5}$-O are deposited with PDB ID 8I41, 8I42, 8I47, 8I48, 8WCQ, 8WCR, and 8JJ3, respectively. Corresponding maps are deposited with EMDB ID EMD-35161, EMD-35162, EMD-35163, EMD-35164, EMD-37446, EMD-37447, and EMD-36339, respectively. The MD data can be downloaded from zenodo https://doi.org/10.5281/zenodo.10792701.

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

## Acknowledgements

We are grateful to the Cryo-Electron Microscopy core facility members at NTU, Singapore. We are also grateful to NISB and NTU for the unconditional access to the central core facility. We thank the CBIS Cryo-Electron Microscopy facility at NUS, Singapore. We thank all the members of the Basak lab and the collaborator lab for editing and critical comments on the manuscript. We also thank Dr. Yvonne Gicheru for editing the manuscript. GLIC plasmid was a generous gift from Dr. Sudha Chakrapani. This work was supported by a start-up grant from SBS, NTU, and the National Research Foundation (NRF) Singapore through an NRF fellowship awarded to S.B. (NRF-NRFF14-2022-0007). D.S. is funded by the UKRI-BBSRC Interdisciplinary Bio-science Doctoral Training Partnership (BB/M011224/1). P.C.B. acknowledges support from the BBSRC (BB/S001247/1). Computing was supported via the Advanced Research Computing facility, Oxford, the ARCHER UK National Super-computing Service, and JADE (EP/T022205/1) granted via the High-End Computing Consortium for Biomolecular Simulation (HECBioSim—http://www.hecbiosim.ac.uk), supported by EPSRC (EP/ R029407/1).

## Author contributions

N.B., Z.L. and S.B. conceived the project and designed the experiments. Z.L. expressed, purified, and reconstituted protein in nanodiscs. D.S. performed molecular dynamic simulations. P.C.B. supervised the MD simulation experiments. A.M.B. purified scaffolding protein MSP1E3D1 and optimized negative stain grids. N.B. and Z.L. made the cryo-EM grids, collected cryo-EM data, and carried out data processing. N.B., Z.L. and S.B. wrote the paper, and all the authors contributed to the discussion.

## Competing interests

The authors declare no competing interests.
