## [Peer Review File · Nature Communications]

Cryo-EM structures of prokaryotic ligand-gated ion channel GLIC provide insights into gating in a lipid environmentREVIEWER COMMENTS

Reviewer #1 (Remarks to the Author):

The manuscript by Bharambe and colleagues describe the structures of a bacterial pentameric ligand gated ion channel GLIC at three different pH using electron cryomicroscopy. The important claim is that the structures are different and more physiologically relevant as the proteins this time were reconstituted in nanodiscs. The structures had overall resolutions in the range of lower 3 Å which allowed to build atomic models and visualize lipids bound to the transmembrane part of the protein. All structures had closed ion pore, molecular dynamics simulations confirmed the absence of ions.

The questions raised in the manuscript are clearly important, and in generally the structures of pentameric ligand-gated ion channels require lipids to adopt the proper fold. However, the results and the conclusions of the manuscript are not ready for publication.

1. Most importantly, the structures at pH 4.0, as well as 5.5 have the closed pore. The authors suggest that the addition of lipids make the structure physiological, however, in order to gain insights into the activation mechanism of gating, the pore has to be open. Otherwise the structure is impossible to reliably interpret.

As a suggestion – the pore can be closed because the open probability of the receptor be “well below 100%” (<https://elifesciences.org/articles/68369>) and the authors could get the open state by potentially collecting more data and classify conformational intermediates. Furthermore, nanodiscs were shown to influence the conformational landscapes of pLGICs (<https://pubmed.ncbi.nlm.nih.gov/33567265/>, <https://www.biorxiv.org/content/10.1101/2022.11.20.517256v1>), perhaps the authors could try other membrane mimetics.

2. The authors claim high overall resolution and the RMSD of atomic models at different pH in the range of 1-1.7 Å. However, the resolution in the ECD is surprisingly poor as can be seen from the validation reports and figures S5, S6. At a resolution of 4 Å the precision of atomic model positioning can be as low as ~1 Å. Therefore, it is not clear how much of a difference is due to the real conformational changes and how much are due to limited resolution. RMSDs were much lower in the TMD, this could be explained by higher resolution in that area. I recommend to get slightly higher resolution for the ECD either by masking, classification or collecting more data.

3. The manuscript is overall poorly written. While this is not the decisive factor on the overall recommendation, the reader only learns that all the structure are closed in the end of the MS.

Furthermore, many important statements are unreferenced, line 110 as an example: “Therefore, the effect of trapped detergent on the pore radius is still under debate”. Line 99: “Most of the existing pLGIC structures are of GLIC”, line 108: “Surprisingly, hydrophobic detergent molecules occlude the pore of the

open state, apparently because of crystallographic artifacts". All those statements have to be either referenced or marked as a hypothesis.

Reviewer #2 (Remarks to the Author):

This paper describes the cryo-EM structures of the bacterial pH-gated ion channel GLIC at different pHs (7.5, 5.5 and 4), where GLIC was first extracted in detergent and then exchanged for nano-discs. In principle, this could lead to new and interesting information, due to the nearly native lipidic environment of the trans-membrane domain. The resolution of the new experimental structures ranges from 3.0 to 3.5 Å, therefore with an exciting potential of revealing new information on the gating mechanism of this ion channel, which has been highly studied in the past 15 years or so but still demands more experimental data.

However, I must say that I strongly disagree with most of the conclusions drawn from these new structures, as discussed below. In addition, the results are not discussed properly in view of the state-of-the-art of the relevant literature, which unfortunately leads to the conclusion that this paper does not meet the required standards for publication.

Main points

-Contrary to what is stated several times in the manuscript, the closed state found in this study is virtually identical to the apo/resting state seen by crystallography (see below)

-Concerning the pH 4 structure (which should be in an open state, although this is a matter of debate, see below), it is well known that indeed the GLIC crystal structure (PDB 3EAM or 4HF1) contains 5 detergent molecules, plus an axial one, which probably maintain the molecule in an artificially widely open state. This is why the original studies submitted this "putatively open state" to long MD simulations, once the detergent molecules in the pore were removed, in a realistic lipidic environment (longer than 1 micro-second). It remained open.

-However, in this study the observed pH 4 state has a pore that is not even open, as confirmed by the pore radius Figure (Figure 4D), the text (p. 9, line 263) and subsequent MD simulations confirm it (line 275). One possibility could be that it could be in a desensitized state, yet it does not have the expected features of the desensitized state (as stated elsewhere in the manuscript, p. 14, line 432).

-Therefore I disagree with the last sentence of the Introduction (p. 5, line 144-45): "Therefore, our multiple novel conformational states of GLIC reconstituted in lipid nanodiscs along with MD simulations data shed light on the gating mechanism of GLIC in a physiologically relevant lipid environment"

-Actually, one possibility that the authors might consider and discuss in the future is the fact that even the situation of GLIC in nanodiscs is not even close to the in vivo one, when GLIC is studied by electrophysiology: indeed, GLIC channels open when an external pH is applied, that is a pH drop on the

ECD side, while the internal pH remains neutral. Obviously, this situation is not reproduced well when all parts of GLIC are exposed to a low pH.

Detailed comments on the manuscript

-p. 4 “Interestingly, all of the deposited GLIC structures were solved in pure or partial detergent environment (26)”: Ref. 26 dates back to 2018, and since then, many more structures of GLIC were deposited in the PDB. Among them, 6ZGD, 6ZGJ and 6ZGK were solved in nanodiscs (Ref. 30 in the paper).

Strikingly, these structures were not used for any detailed comparison with the structures of this paper, no Rmsd between them is reported and no proper discussion is included as to why the two sets of GLIC cryo-EM structures in nano-discs differ so much (the present reviewer is not a co-author of Ref. 30 and is, in this matter, completely agnostic).

-p. 5 Why are all the results already described and discussed at the end of the Introduction?

-p. 5 “The crystal structures have lower number of lipid molecules” Please explain: a low number?

State observed at pH 7

-p. 6 Comparison of the two apo-structures of this paper (aso-pH7.5 and lipids-pH7.5) have an rmsd of 1.1 Å, while the rmsd with the crystal resting/apo state is 1.7 Å (PDB 4NPQ).

First of all, 4NPQ contains 4 different pentamers: what is the rmsd of GLIC-lipids-pH7.5 with each one of the 4 pentamers? This would at least give an idea of the variability of this number. Second, why is rmsd=1.7 Å considered to be due to two different structures while 1.1 Å is not? What is the limit? Could it be that 1.7 Å is well within the error possibly induced by crystal contacts in the ECD domain? Third, a visual inspection of the transmembrane domain superposed to each one of the 4 pentamers of 4NPQ shows that they are virtually identical (confirmed by Fig. 2F). At the level of the ECD, the differences seem mostly due to packing interactions and I doubt very much they can lead to the assignment of this apo-states as “new structures”.

In other words, by visual inspection of the data and by analyzing the Figures provided here I strongly disagree with the following sentence in the manuscript:

-Summary p.2 “whereas the extracellular domain (ECD) adopts a novel conformation”

which is propagated in several places of the manuscript:

-p. 5 (line 130): “represent a closed state that is distinct from the existing apo state in detergent”

-p. 11 line 348: “The overall conformation of GLIC-lipids-pH7.5 differs from the apo state in multiple regions”: No

State observed at pH 4

-p. 2 The protonated state is described to adopt “a conformation unlike any existing protonated state” in the Abstract, but the significance of the observation that its pore is actually closed (contrary to what is expected) is not explained.

-p. 13-14. It is clear that the authors do not know what functional state corresponds to this structure, thereby considerably lowering the potential interest of this manuscript.

Minor points

-Some formatting problems with Refs #45 (line 730), #57 (line 766) and consistency among them (stop after the doi reference?)

-Incidentally, contrary to what is stated p. 4 (line 109), the binding of detergent molecules in the pore of the pH 4 crystal structure of GLIC should not be described as a crystallographic artefact, but rather a solubilization artefact: a crystallization artefact would involve distortion of the structure of GLIC by protein-protein contacts due to crystal packing inherent to the crystallization – which actually are present, but not in the pore itself.

Reviewer #3 (Remarks to the Author):

This manuscript investigates the structure of the prokaryotic ligand gated ion channel, GLIC, reconstituted in lipid nanodiscs. The team set out to probe the GLIC gating cycle and to compare with detergent based preparations that have previously yielded a somewhat confusing array of structures. They present four new structures of GLIC in two different lipid mixtures at pH 7.5, 5.5, and 4.0. First, they find that the two lipid-mixtures cause differences in protein structure at neutral pH. They then explore the various closed states stabilized at pH 7.5, 5.5, and 4.0 (functionally annotated by MD simulations), that differ from their detergent counterparts, and all share a constriction at 9'1. Finally, they show that lipids bind GLIC in a state dependent manner suggesting a role of lipids in receptor gating. Overall, the quality of the underlying structural data appear solid. Perceived weaknesses include lack of a clear conceptual advance, overstatement of novelty and significance, and unpolished figures and text. GLIC has been very useful as a model system for testing aspects of pLGIC structure-function that were not yet tractable in eukaryotic receptors. Many groups have now published substantially higher resolution structures of eukaryotic pLGICs than are found here, in lipid nanodiscs, in multiple conformational states, with good physiology and MD to accompany them. Accordingly, studies on GLIC structure-function will likely be of interest only to specialists, and not to the broad readership of Nature Communications.

Major points:

- The new structural information does not clarify where the earlier structures sit in the gating cycle or which ones are not relevant. The new structures are somewhat ambiguously assigned to resting-like states.
- Overstating the conceptual advance and inaccurate descriptions of the existing literature. For example, the introduction states “so far there is no unequivocal assignment of the functional state to a specific conformation.” While it is true that there are disconnects in many cases between what one sees in a pLGIC structure and what one might expect from physiology, there are many structures for which there is consensus agreement on them representing a resting or a desensitized state. This statement also implies that the author’s new study solves this problem of no structures being assigned to states with confidence yet. It is not clear why the new structures might be more reliably assigned to a functional state than all pre-existing structures.
- The title suggests that the findings “reveal” the gating mechanism but the team only observes resting-like states.
- In methods for data processing there is no mention of 3D classification, which appears in processing figures.
- “all of our nanodisc reconstituted structures adopt novel conformational states” is confusing. A novel observation should be tested. How do we know that these new states are not simply nanodisc dependent?
- Related to the above comment/question, assertions that the nanodisc structures are better representative of ‘real’ conformations seen physiologically needs to be better supported. Nanodiscs can themselves introduce structural artifacts.

Minor points:

- The authors should specifically mention that it is a pH sensitive channel in the abstract
- The authors mention that it is “surprising” that there is state-dependent lipid interaction: then later say as seen before (line 143). Also should cite PMID: 36251999
- Line 76 “the optimal activation...” sentence is not needed
- Missing a word in line 113
- Capitalization of “supplementary figure” is inconsistent
- Rerun hole but change the maximum diameter (to get it to run all the way to the top) or just show the TMD region
- Go back on forth with pH 7 or pH 7.5: 7.0 line 329

- Lines 328-332 are repetitive
- Typo in 431: conformation
- Typo 436: simulations
- Typo 492: was should be were
- Change the axes in Figure 4D. really difficult to see
- Figure 1: having both models and maps here doesn't really provide any additional information
- Many figures appear as rough drafts with mixed fonts, partial rendering, missing labels.

Reviewers' comments:

We thank all three reviewers for their time to review our manuscript in-depth and for the insightful comments. We have incorporated all the suggested changes. We hope the reviewers will find the revision satisfactory and the manuscript acceptable for publication.

Reviewer #1 (Remarks to the Author):

The manuscript by Bharambe and colleagues describe the structures of a bacterial pentameric ligand gated ion channel GLIC at three different pH using electron cryomicroscopy. The important claim is that the structures are different and more physiologically relevant as the proteins this time were reconstituted in nanodiscs. The structures had overall resolutions in the range of lower 3 Å which allowed to build atomic models and visualize lipids bound to the transmembrane part of the protein. All structures had closed ion pore, molecular dynamics simulations confirmed the absence of ions.

The questions raised in the manuscript are clearly important, and in generally the structures of pentameric ligand-gated ion channels require lipids to adopt the proper fold. However, the results and the conclusions of the manuscript are not ready for publication.

1. Most importantly, the structures at pH 4.0, as well as 5.5 have the closed pore. The authors suggest that the addition of lipids make the structure physiological, however, in order to gain insights into the activation mechanism of gating, the pore has to be open. Otherwise the structure is impossible to reliably interpret.

As a suggestion – the pore can be closed because the open probability of the receptor be “well below 100%” (<https://elifesciences.org/articles/68369>) and the authors could get the open state by potentially collecting more data and classify conformational intermediates. Furthermore, nanodiscs were shown to influence the conformational landscapes of pLGICs (<https://pubmed.ncbi.nlm.nih.gov/33567265/>, <https://www.biorxiv.org/content/10.1101/2022.11.20.517256v1>), perhaps the authors could try other membrane mimetics.

Thank you for your critical comments. As per your suggestion, we have now recollected and reprocessed the data, which significantly improved the map quality, as reflected by the increase in overall and local resolution. Most importantly, we have now solved an open state of GLIC with a wider and hydrated pore, which is further confirmed by MD simulations. We also identified more intermediate states. We have built more accurate models and compared them in detail in the main manuscript. We built more lipid molecules located in previously unseen positions and discussed their importance in gating. Below, we have listed the major changes in the cryo-EM data quality.

For GLIC-*aso*-pH7.5 (Now it is called GLIC_{aso}-pH7.5c): We have reprocessed the data and the nominal resolution improved moderately from 3.53 Å to 3.42 Å.

For GLIC-lipids-pH7.5 (Now it is called GLIC_{lipids}-pH7.5c): We have recollected a large dataset (~14,000 movies) using Falcon 4i with selectris X energy filter (Thermo fisher). We performed extensive data processing using both cryoSPARC and RELION. The nominal resolution improved from 3.01 Å to 2.92 Å with significant improvement of local resolution, especially in the ECD (Supplementary figure 2g). This helped to build more accurate model.

For GLIC-lipids-pH5.5 (Now it is called GLIC_{lipids}-pH5.5c): We have reprocessed the data. We observed significant improvement in the quality of the final map mainly because of the use of TOPAZ picking, which improves the selection of good quality particles. There was preferred orientations issue with the protein particles on the cryo-EM grid. During data processing, we identified and removed particles that dominated the bottom view and interfered with particle alignment. After thorough data

processing, the resolution improved from 3.34 Å to 2.70 Å. We have included detailed data processing information in the Methods section and in the data processing flow chart (Supplementary figure 4).

For GLIC-lipids-pH4.0 (Now it is called GLIC_{lipids}-pH4.0_{C1 or C2 or O}): We have recollected a large dataset (~14,000 movies) using Falcon 4i with selectris X energy filter (Thermo Fisher). We performed similar extensive data processing using both cryoSPARC and RELION. This extensive data processing led to the identification of three different structures (C1, C2 and O) from a single sample in nominal resolution ranges of 2.65 Å -3.35 Å. Among the three structures, GLIC_{lipids}-pH4.0_O (35% of total population) has a wider and hydrated pore and represents an open state.

For GLIC_{lipids}-pH2.5_O: We collected the dataset using K3 detector with BioQuantum energy filter. Extensive data processing yielded nominal resolution of 2.65 Å. The conformation of this state is similar to the GLIC_{lipids}-pH4.0_O state with backbone (C α) RMSD of 0.20 Å. At this pH, the open probability increased to 100% in our experiment.

In the case of the glycine receptor, the authors speculated that either lipids or nanodiscs might influence the equilibrium of activation and desensitization (PMID: 33567265). Interestingly, the lipid compositions were different in the nanodisc and SMA samples, which could be the main reason for the equilibrium shift. In this study, after new data collection and processing, we determined the open conformation at pH 4.0. At pH 2.5, the open probability increased to 100%. Interestingly, the pH₅₀ of liposome-reconstituted GLIC was calculated to be 2.9. This clearly supports our data that we see an increase in open probability below pH 2.9.

In the case of ELIC, the structure was captured in the pre-active conformation in smaller nanodiscs. However, in this study, we observed several conformational states along with an open conformation. Nevertheless, in our revised manuscript, we have included the limitations of our study and future directions, such as the determination of the structure of GLIC in liposomes.

2. The authors claim high overall resolution and the RMSD of atomic models at different pH in the range of 1-1.7 Å. However, the resolution in the ECD is surprisingly poor as can be seen from the validation reports and figures S5, S6. At a resolution of 4 Å the precision of atomic model positioning can be as low as ~1 Å. Therefore, it is not clear how much of a difference is due to the real conformational changes and how much are due to limited resolution. RMSDs were much lower in the TMD, this could be explained by higher resolution in that area. I recommend to get slightly higher resolution for the ECD either by masking, classification or collecting more data.

Thank you for your suggestions. The resolutions (both nominal and local) are now significantly improved after recollected and reprocessing the data. We have now built more precise models in both ECD and TMD. For comparison, we have now calculated the RMSD of the backbone (C α), which provides information more accurately on conformational changes in various structures (Supplementary fig. 3a,b and 10). The details are included in the main text.

3. The manuscript is overall poorly written. While this is not the decisive factor on the overall recommendation, the reader only learns that all the structure are closed in the end of the MS. Furthermore, many important statements are unreferenced, line 110 as an example: "Therefore, the effect of trapped detergent on the pore radius is still under debate". Line 99: "Most of the existing pLGIC structures are of GLIC", line 108:" Surprisingly, hydrophobic detergent molecules occlude the pore of the open state, apparently because of crystallographic artifacts". All those statements have to be either referenced or marked as a hypothesis.

We have now rephrased several sentences and provided references for important statements.

Reviewer #2 (Remarks to the Author)

This paper describes the cryo-EM structures of the bacterial pH-gated ion channel GLIC at different pHs (7.5, 5.5 and 4), where GLIC was first extracted in detergent and then exchanged for nano-discs. In principle, this could lead to new and interesting information, due to the nearly native lipidic environment of the trans-membrane domain. The resolution of the new experimental structures ranges from 3.0 to 3.5 Å, therefore with an exciting potential of revealing new information on the gating mechanism of this ion channel, which has been highly studied in the past 15 years or so but still demands more experimental data. However, I must say that I strongly disagree with most of the conclusions drawn from these new structures, as discussed below. In addition, the results are not discussed properly in view of the state-of-the-art of the relevant literature, which unfortunately leads to the conclusion that this paper does not meet the required standards for publication.

Thank you for your comments on the new experimental structures of GLIC. Below, we have answered your questions point-by-point to clarify your concerns.

Main points

-Contrary to what is stated several times in the manuscript, the closed state found in this study is virtually identical to the apo/resting state seen by crystallography (see below)

*Thank you for your comments and concerns. Our apo structure is indeed a state different from the existing apo state. First to clarify that GLIC was solubilized in **detergent** at pH 7.0 and the structure was solved by cryo-EM (6ZGD). Therefore, 6ZGD (pH7.0) was **not** a nanodisc-reconstituted GLIC structure. In the current manuscript, we have used the GLIC apo structure for comparison, which has been solved by X-ray crystallography (PDB ID: 4NPQ). There are four pentamers in the asymmetric unit. Apparently, the crystal contacts have minimal effects on the conformations as all four structures are similar with backbone (Ca) RMSD ≤ 0.36 Å. The overall conformation of TMD in our apo state is similar to the existing apo state (4NPQ). However, the ECD in 4NPQ is more compacted compared to our apo state (Accessible surface area from 54,089 Å² (our apo-ECD) to 48,123 Å² (4NPQ-ECD)). This could be the effect of crystallization conditions. However, the accessible surface area (ASA) of the GLIC apo structure solved by cryo-EM (PDB ID: 6ZGD) is 47,207 Å² (other two closed structures have similar ASA), which is similar to the 4NPQ structure. Therefore, crystallization conditions may have a minimal effect on ECD compaction in the case of GLIC. In addition, we observed clockwise rotation in our ECD compared to 4NPQ (Figure 1d). Compared to our apo state, the backbone (Ca) RMSD of ECD with 4NPQ and 6ZGD is 2.7 Å and 3.5 Å, respectively. Therefore, the detergent environment may have stabilized the closed states in 4NPQ and 6ZGD, which are distinct from our apo state. Hence, the new conformation of our nanodisc-reconstituted GLIC captured at pH 7.5 is due to the lipid environment.*

-Concerning the pH 4 structure (which should be in an open state, although this is a matter of debate, see below), it is well known that indeed the GLIC crystal structure (PDB 3EAM or 4HFI) contains 5 detergent molecules, plus an axial one, which probably maintain the molecule in an artificially widely open state. This is why the original studies submitted this “putatively open state” to long MD

simulations, once the detergent molecules in the pore were removed, in a realistic lipidic environment (longer than 1 micro-second). It remained open.

-However, in this study the observed pH 4 state has a pore that is not even open, as confirmed by the pore radius Figure (Figure 4D), the text (p. 9, line 263) and subsequent MD simulations confirm it (line 275). One possibility could be that it could be in a desensitized state, yet it does not have the expected features of the desensitized state (as stated elsewhere in the manuscript, p. 14, line 432).
-Therefore I disagree with the last sentence of the Introduction (p. 5, line 144-45): “Therefore, our multiple novel conformational states of GLIC reconstituted in lipid nanodiscs along with MD simulations data shed light on the gating mechanism of GLIC in a physiologically relevant lipid environment”

Thank you for this intriguing question. We recollected a large set of cryo-EM data (~14,000 movies) of nanodisc-reconstituted GLIC at pH 4.0 using Falcon 4i with selectris X energy filter (Thermo Fisher). We performed extensive data processing using cryoSPARC and RELION. After careful data processing, we identified three different 3D classes. After 3D refinement, we identified three unique conformations (C1, C2 and O states), in the resolution range of 2.65-3.35 Å. Among these three structures, GLIC_{lipids-pH4.0_O} or O state adopts a conformation with compacted and counter-clockwise twisted ECD compared to our apo state. The pore is wider and contain several water molecules. The radius of the pore at the hydrophobic gate (I9) is wider than a hydrated Na⁺ ion. MD simulations also revealed that the pore is hydrated, suggesting that the O state represents an open state. We solved the cryo-EM structure of nanodisc-reconstituted GLIC at pH 2.5 with a nominal resolution of 2.65 Å with an overlapping conformation to GLIC_{lipids-pH4.0_O} state. Surprisingly, the population of GLIC_{lipids-pH4.0_O} is only 35% of the total intact protein particles on the cryo-EM grid. Reducing the pH to 2.5 increased the open probability to 100% in our experiment, which agrees with the pH₅₀ (2.9) of liposome-reconstituted GLIC. Therefore, our new structures (open and several intermediate states mentioned in the main text) justify our last sentence in the introduction.

-Actually, one possibility that the authors might consider and discuss in the future is the fact that even the situation of GLIC in nanodiscs is not even close to the in vivo one, when GLIC is studied by electrophysiology: indeed, GLIC channels open when an external pH is applied, that is a pH drop on the ECD side, while the internal pH remains neutral. Obviously, this situation is not reproduced well when all parts of GLIC are exposed to a low pH.

Thank you for this suggestion. Now we can see that the channel can be activated in the lipid nanodisc at acidic pH. However, we agree with the reviewer, and this analogy applies to all existing GLIC structures as well. However, a previous study showed that a low pH on the TMD has a minimal effect on channel activation. Therefore, in terms of functionality, we assume a minimal effect of low pH on the TMD side (PMID: 29281623). We have now mentioned these statements in the limitation section of the main text with reference (“However, a previous study showed that the low pH has a minimal effect on the TMD in activating the channel”). We have also included future directions, such as determination of the structure of GLIC in liposomes, to better understand the gating mechanism.

Detailed comments on the manuscript

-p. 4 “Interestingly, all of the deposited GLIC structures were solved in pure or partial detergent environment (26)”: Ref. 26 dates back to 2018, and since then, many more structures of GLIC were

deposited in the PDB. Among them, 6ZGD, 6ZGJ and 6ZGK were solved in nanodiscs (Ref. 30 in the paper).

Strikingly, these structures were not used for any detailed comparison with the structures of this paper, no Rmsd between them is reported and no proper discussion is included as to why the two sets of GLIC cryo-EM structures in nano-discs differ so much (the present reviewer is not a co-author of Ref. 30 and is, in this matter, completely agnostic).

*Thank you for bringing this point up. Most recent studies on GLIC have mainly focused on either identifying the residues responsible for proton sensors or gaining a better understanding of the effect of modulators, which we have now mentioned in the main text with references. The overall conformations of most of these structures are similar to known existing conformations. A breakthrough study on GLIC by Rovšnik et al. showed that GLIC conformations can be captured by cryo-EM. However, note that GLIC **has not** been reconstituted in the nanodisc. It is still in **detergent (DDM) environments (6ZGD, 6ZGJ and 6ZGK)**. Therefore, the conformation of their apo state is close to the existing apo state of GLIC solved by X-ray crystallography and differs from our apo state. We have now compared the 6ZGD with our apo state and included it in the main text (Supplementary figure 3c). Recently, the same group reprocessed the data and found that a very small population of particles yielded an apo state with bound lipid molecules. We have also mentioned this in the main text.*

-p. 5 Why are all the results already described and discussed at the end of the Introduction?

Thank you for this valuable comment. We have now rewritten this section.

-p. 5 “The crystal structures have lower number of lipid molecules” Please explain: a low number?

We have now included the number of lipid binding sites identified by X-ray crystallography. The following sentence has been added to the main text.

“Only three unique lipid binding sites have been identified in GLIC so far with high confidence using X-ray crystallography.”

The new cryo-EM structure of detergent-solubilized GLIC (reprocessed data) has five lipids. This information has also been included in the main text. The following sentence has been added to the main text.

“In the recently solved cryo-EM structure of detergent solubilized GLIC, two lipids in the upper leaflet and three lipids in the lower leaflet were observed in each subunit...”

State observed at pH 7

-p. 6 Comparison of the two apo-structures of this paper (aso-pH7.5 and lipids-pH7.5) have an rmsd of 1.1 Å, while the rmsd with the crystal resting/apo state is 1.7 Å (PDB 4NPQ).

First of all, 4NPQ contains 4 different pentamers: what is the rmsd of GLIC-lipids-pH7.5 with each one of the 4 pentamers? This would at least give an idea of the variability of this number. Second, why is rmsd=1.7 Å considered to be due to two different structures while 1.1 Å is not? What is the limit? Could it be that 1.7 Å is well within the error possibly induced by crystal contacts in the ECD domain? Third, a visual inspection of the transmembrane domain superposed to each one of the 4 pentamers of 4NPQ shows that they are virtually identical (confirmed by Fig. 2F). At the level of the ECD, the differences seem mostly due to packing interactions and I doubt very much they can lead to the assignment of this apo-states as “new structures”.

Thank you for pointing this out. The resolution and quality of the cryo-EM maps have significantly improved, which has increased the accuracy of the models. Below are the details.

For GLIC-*aso*-pH7.5 (Now it is called GLIC_{*aso*}-pH7.5_{*c*}): We have reprocessed the data and the nominal resolution improved moderately from 3.53 Å to 3.42 Å.

For GLIC-lipids-pH7.5 (Now it is called GLIC_{lipids}-pH7.5_{*c*}): We have recollected a large dataset (~14,000 movies) using Falcon 4i with selectris X energy filter (Thermo fisher). We performed extensive data processing using both cryoSPARC and RELION. The nominal resolution improved from 3.01 Å to 2.92 Å with significant improvement of local resolution, especially in the ECD (Supplementary figure 2g). This helped to build more accurate model.

For GLIC-lipids-pH5.5 (Now it is called GLIC_{lipids}-pH5.5_{*c*}): We have reprocessed the data. We observed significant improvement in the quality of the final map mainly because of the use of TOPAZ picking, which improves the selection of good quality particles. There was preferred orientations issue with the protein particles on the cryo-EM grid. During data processing, we identified and removed particles that dominated the bottom view and interfered with particle alignment. After thorough data processing, the resolution improved from 3.34 Å to 2.70 Å. We have included detailed data processing information in the Methods section and in the data processing flow chart (Supplementary figure 4).

For GLIC-lipids-pH4.0 (Now it is called GLIC_{lipids}-pH4.0_{*C1* or *C2* or *O*}): We have recollected a large dataset (~14,000 movies) using Falcon 4i with selectris X energy filter (Thermo Fisher). We performed similar extensive data processing using both cryoSPARC and RELION. This extensive data processing led to the identification of three different structures (*C1*, *C2* and *O*) from a single sample in nominal resolution ranges of 2.65 Å -3.35 Å. Among the three structures, GLIC_{lipids}-pH4.0_{*O*} (35% of total population) has a wider and hydrated pore and represents an open state.

For GLIC_{lipids}-pH2.5_{*o*}: We collected the dataset using K3 detector with BioQuantum energy filter. Extensive data processing yielded nominal resolution of 2.65 Å. The conformation of this state is similar to the GLIC_{lipids}-pH4.0_{*O*} state with backbone (*Ca*) RMSD of 0.20 Å. At this pH, the open probability increased to 100% in our experiment.

As it is evident that the overall resolution improved from the previous submission, this helps us to compare the structures more accurately. For structural comparison, we calculated the RMSD of the backbone (*Ca*). Four pentamers present in the asymmetric unit of 4NPQ have backbone RMSD ≤ 0.36 Å indicating very similar conformations. Therefore, we used a single pentamer for comparison. The backbone RMSD in the ECD of our GLIC_{*aso*}-pH7.5_{*c*} and GLIC_{lipids}-pH7.5_{*c*} is 0.52 Å. However, the backbone RMSD in the ECD of GLIC_{lipids}-pH7.5_{*c*} and GLIC_{detergent}-pH7.5_{*c*} (4NPQ) is 2.7 Å. This clearly reflects the conformational differences between these structures in the ECD. The plausible reasons for these differences are mentioned in one of the previous answers (First answer of the main points). Notably, the major conformational changes observed in our apo structure are at $\beta 1$ - $\beta 2$, loop F, loop A, and loop C, which are located away from the crystal contacts, indicating the minimum effect of crystal contacts in this region.

In other words, by visual inspection of the data and by analyzing the Figures provided here I strongly disagree with the following sentence in the manuscript:

-Summary p.2 “whereas the extracellular domain (ECD) adopts a novel conformation”

which is propagated in several places of the manuscript:

-p. 5 (line 130): “represent a closed state that is distinct from the existing apo state in detergent”

-p. 11 line 348: ‘The overall conformation of GLIC-lipids-pH7.5 differs from the apo state in multiple regions’: No

The newly processed data yielded a higher resolution map with improved local resolution, which helped to build and compare the models more accurately. The clear conformational changes are now highlighted in figure 1d-i. Therefore, our apo state truly represents an unprecedented state close to the resting state (Fig. 7).

State observed at pH 4

-p. 2 The protonated state is described to adopt “a conformation unlike any existing protonated state” in the Abstract, but the significance of the observation that its pore is actually closed (contrary to what is expected) is not explained.

-p. 13-14. It is clear that the authors do not know what functional state corresponds to this structure, thereby considerably lowering the potential interest of this manuscript.

This is a valid concern. We have now determined the high-resolution structures of nanodisc-reconstituted GLIC with compacted and twisted ECD along with wider pore (at pH 4.0 and 2.5). The pore is hydrated, which is evident from several water molecules in the cryo-EM map and MD simulations. Therefore, both GLIC_{lipids-pH4.0o} and GLIC_{lipids-pH2.5o} represent an open state. In addition, we have now solved two intermediate states C1 and C2 at pH 4.0, where C2 likely represents a pre-open state.

Minor points

-Some formatting problems with Refs #45 (line 730), #57 (line 766) and consistency among them (stop after the doi reference?)

We have now fixed the references.

-Incidentally, contrary to what is stated p. 4 (line 109), the binding of detergent molecules in the pore of the pH 4 crystal structure of GLIC should not be described as a crystallographic artefact, but rather a solubilization artefact: a crystallization artefact would involve distortion of the structure of GLIC by protein-protein contacts due to crystal packing inherent to the crystallization – which actually are present, but not in the pore itself.

We have now removed this statement.

Reviewer #3 (Remarks to the Author):

This manuscript investigates the structure of the prokaryotic ligand gated ion channel, GLIC, reconstituted in lipid nanodiscs. The team set out to probe the GLIC gating cycle and to compare with detergent based preparations that have previously yielded a somewhat confusing array of structures. They present four new structures of GLIC in two different lipid mixtures at pH 7.5, 5.5, and 4.0. First, they find that the two lipid-mixtures cause differences in protein structure at neutral pH. They then explore the various closed states stabilized at pH 7.5, 5.5, and 4.0 (functionally annotated by MD simulations), that differ from their detergent counterparts, and all share a constriction at 9°I. Finally, they show that lipids bind GLIC in a state dependent manner suggesting a role of lipids in receptor gating. Overall, the quality of the underlying structural data appear solid. Perceived weaknesses include lack of a clear conceptual advance, overstatement of novelty and significance, and unpolished figures and text. GLIC has been very useful as a model system for testing aspects of pLGIC structure-

function that were not yet tractable in eukaryotic receptors. Many groups have now published substantially higher resolution structures of eukaryotic pLGICs than are found here, in lipid nanodiscs, in multiple conformational states, with good physiology and MD to accompany them. Accordingly, studies on GLIC structure-function will likely be of interest only to specialists, and not to the broad readership of Nature Communications.

Thank you for your valuable comments. Below, we have answered your questions point-by-point to clarify your concerns.

Major points:

- The new structural information does not clarify where the earlier structures sit in the gating cycle or which ones are not relevant. The new structures are somewhat ambiguously assigned to resting-like states.

We have reprocessed and recollected data that improved the resolution significantly. More importantly, we identified new intermediate states along with an open state. The structures show a cascade of conformational changes along the activation pathway upon reducing the pH to acidic conditions. We compared our structures with existing crystal structures. Based on the conformational changes, we have depicted a structural flowchart for gating in figure 7.

- Overstating the conceptual advance and inaccurate descriptions of the existing literature. For example, the introduction states “so far there is no unequivocal assignment of the functional state to a specific conformation.” While it is true that there are disconnects in many cases between what one sees in a pLGIC structure and what one might expect from physiology, there are many structures for which there is consensus agreement on them representing a resting or a desensitized state. This statement also implies that the author’s new study solves this problem of no structures being assigned to states with confidence yet. It is not clear why the new structures might be more reliably assigned to a functional state than all pre-existing structures.

We have rephrased this sentence. It states, “Although significant research has been conducted on the conformational states of the pLGIC family, questions remain on the unequivocal assignment of several functional states to a specific conformation, especially in prokaryotic ligand-gated ion channels”.

We have solved new structures along with an open state. We observed several lipids in the TMD and noticed that there are state-dependent interactions with protein. Based on the conformational changes and specific state-dependent lipid interactions, we attempted to assign our state along the activation pathway (Figure 7).

- The title suggests that the findings “reveal” the gating mechanism but the team only observes resting-like states.

This is now justified because we have identified several intermediate states along with an open conformation with hydrated pore. This time, we observed extra lipid molecules compared to the last submission. Among several lipids observed in the TMD, lipid 9 is found in the outer leaflet of the C2 and O states and involves polar interactions with neighbouring key residues present in the M2–M3 loop and pre-M1. Similarly, at acidic pH, the acyl chain of lipid 2 wedges in the cavity formed by M1, M3, and M4 helices, which has been shown to be an important region for channel gating. Therefore, together with MD simulations, our cryo-EM structures reveal previously unseen conformations of GLIC in the lipid environment and provide comprehensive insights into the importance of lipids on gating.

- In methods for data processing there is no mention of 3D classification, which appears in processing figures.

We have now included this in the revised manuscript.

- “all of our nanodisc reconstituted structures adopt novel conformational states” is confusing. A novel observation should be tested. How do we know that these new states are not simply nanodisc dependent?

We have toned down and removed “novel” statement from the main text.

The maximum diagonal distance of M4 in the TMD is measured below 75 Å in existing high-resolution structures of the GLIC. In the membrane-embedded GLIC, the maximum diameter of the TMD was measured to be ~75 Å (PMID: 30181288). Therefore, for our current structural investigation, we used the scaffolding protein MSP1E3D1, which forms nanodiscs with a maximum diameter of 129 Å, making it suitable for our study. We have included these statements in the discussion section of the main text.

In addition, the pH_{50} of liposome-reconstituted GLIC was calculated to be 2.9 (PMID: 22474322). In our experiment, we observed a 35% open population at pH 4.0. At pH 2.5, the open probability increased to 100%. This clearly supports our data that we see an increase in open probability below pH 2.9. Therefore, we expect no effect of nanodiscs on the specific conformational state of GLIC.

- Related to the above comment/question, assertions that the nanodisc structures are better representative of ‘real’ conformations seen physiologically needs to be better supported. Nanodiscs can themselves introduce structural artifacts.

We have now solved an open conformation of GLIC. Structural comparison among our structures revealed systematic conformational changes in the ECD and pore of the TMD. We also identified several lipids that interact with proteins in a state-dependent manner, which was predicted earlier using MD simulations. The open probability of GLIC correlates with the previous patch-clamp study of liposome-reconstituted GLIC (as mentioned before). Therefore, we could hypothesize that the effect of nanodiscs on stabilizing a specific non-physiological state of GLIC is none. Nevertheless, we have mentioned in the limitation section that “determination of the structure of GLIC in a more native-like environment, such as liposomes, would provide further insights into the gating mechanism at near-native environment”.

Minor points:

- The authors should specifically mention that it is a pH sensitive channel in the abstract

Done. The sentence is now rephrased to “GLIC, a proton-activated prokaryotic ligand-gated ion channel, has profound structural and functional similarities with its eukaryotic counterparts...”.

- The authors mention that it is “surprising” that there is state-dependent lipid interaction: then later say as seen before (line 143). Also should cite PMID: 36251999

We apologize for the confusion. With that reference, we wanted to state that it was predicted using MD simulations. In our current study, we observed state-dependent protein-lipid interactions using cryo-EM experiments. We have now rephrased the sentence as follows: “While the position and interaction patterns of lipids at the lower leaflet are conserved, few lipids at the upper leaflet have positional fluctuations in specific states with differential interaction networks, indicating a state-dependent role, as previously predicted in other pLGICs using MD simulations”.

- Line 76 “the optimal activation...” sentence is not needed

The sentence has been removed.

- Missing a word in line 113

The sentence has been rephrased as follows: “However, the recent cryo-EM structures of detergent solubilized GLIC were solved at various pH conditions (pH 7.0, 5.0 and 3.0), which appeared to be in a closed state but distinct from the known GLIC structures”.

- Capitalization of “supplementary figure” is inconsistent

We have fixed this inconsistency in the revised manuscript.

- Rerun hole but change the maximum diameter (to get it to run all the way to the top) or just show the TMD region

In the revised manuscript, we have shown only the TMD region.

- Go back on forth with pH 7 or pH 7.5: 7.0 line 329

We have fixed this .

- Lines 328-332 are repetitive

These sentences have been removed .

- Typo in 431: conformation

Fixed

- Typo 436: simulations

Fixed

- Typo 492: was should be were

Fixed now.

- Change the axes in Figure 4D. really difficult to see

We have taken this into account. In the revised manuscript, all the figures have been re-made.

- Figure 1: having both models and maps here doesn't really provide any additional information

All figures have been re-made.

- Many figures appear as rough drafts with mixed fonts, partial rendering, missing labels.

We have taken this into account. In the revised manuscript, all the figures have been revised.

REVIEWER COMMENTS

Reviewer #1 (Remarks to the Author):

The manuscript by Bharambe and colleagues has significantly improved, revealing more structural intermediates of the activation cycle of GLIC at an enhanced resolution. The authors have now demonstrated a wider pore in one of the structures at pH 4 and 2.5. Consequently, they could align the structures in an apparent activation sequence. The manuscript is also much more readable and has better references to the literature.

However, there are some serious concerns that need to be addressed before the manuscript can be published.

Major Concerns:

1. My primary concern is the assignment of the structures to their presumed functional roles. GLIC may desensitize even without cholesterol, and the structures of the desensitized receptors will likely look more similar to the open state than to the closed ones and will be unable to pass ions. In the current manuscript, the justification for assigning the structure to intermediate and pre-open states is unclear and requires further work.
2. On line 215, the authors built water molecules into the EM maps at a resolution of 2.5-2.7 Å. I have serious concerns about whether this can be reliably done. In fact, publications suggest that reliable building of water molecules can be done at a resolution better than 2 Å (PMID: 34473085). Thorough validation would be required to make placement of water in such maps close to the symmetry axis (like in the pore in Figure 4g) reliable. I did not find such validation in the manuscript text.
3. The fact that MD simulations showed good hydration of the pore (line 328) by water molecules does not necessarily mean that it is ion-permeable. The authors base their assignment of GLIClipids-331 pH4.00 as an open state; however, more evidence is required that this state allows ion passage.
4. The authors describe the interactions of lipids with the channel; however, they do not seem to draw any conclusions from these interactions. A series of point mutations on GLIC could answer these questions. Some mutations have been reported in the literature and are discussed in the text; others could be performed based on new structural findings.

5. For the next round of reviews, it would be important to provide validation reports from depositions of structures to publicly accessible databases.

Minor Concerns:

1. The statement that reported apo-structures are in previously unseen conformations (line 194) is obvious and somewhat redundant as it is described in the preceding paragraph. Should this be removed or discussed in more detail?
2. The authors report a second digit at resolutions of 2.7 Å. How relevant is this?
3. I am unsure what the authors mean by “holistic glimpse” (line 580).

Reviewer #2 (Remarks to the Author):

Comments on the revised manuscript of Basak et al. Nature Commun. Oct 2023

Using cryo-EM the authors recollected more data of GLIC reconstituted in a nanopore and obtained a continuum of structures of very good quality between the closed form and the open form, in a lipidic environment, by varying the pH.

They describe very interesting new experimental structures, especially for the channel of the open form with very nice density for several water pentagons in the pore as shown in Fig. 13h.

I still have important concerns to share about the present state of the manuscript.

In particular a number of important papers relevant to the present manuscript are not discussed properly.

-Position of lipids

The paper in eLife by Lindahl and coll. (Ref. 47) should be discussed more fully, as it reports also the position of the lipids for at least one form of GLIC, and this is one of the main points of the manuscript. Their PDB structure 8ATG is in fact publicly available since Sept. 6th.

The authors should use it and compare the positions of lipids observed here to those they observe. See Fig. 14 and 15, which should be changed.

p. 11 line 347 “However, a comparison of the lipids was not feasible due to unavailability of the model, at the time of manuscript preparation...”

p. 17 line 538 “(PDB is not available)”: In fact, it is available.

Their full MD trajectories are also actually available since April 25th.

In fact, the statement line 580

‘In summary, our structures provide the first holistic glimpse of the gating mechanism and state-dependent protein-lipid interactions in GLIC’

should be toned down as there has been a number of studies on the gating mechanism of GLIC (Ref 9), as well as on “state-dependent protein-lipid interactions (Ref 47, Figure 4).

-Diameter of the pore

Figure 5: The reported diameter of the water molecule (dotted line) seems to be wrong: it should be 2.78-2.8 Angstrom, not 2.5 A. See also p. 33 the Legend of the Figure 5.

-Scenario for ion channel gating and ion permeation

How does the scenario described in Figure 7 and in the Discussion compare with the experimental work of Menny et al. 2017? See <https://pubmed.ncbi.nlm.nih.gov/28294942/>

How does it compare with T. Allen and coll. article on GLIC transition studied by the string method in MD simulations in 2017? See <https://pubmed.ncbi.nlm.nih.gov/28487483/>

In particular, the transition described in this PNAS paper is not symmetrical, each monomer undergoing a transition in a progressive manner. In contrast, all structures presented here have C5 symmetry by construction. Is this an inevitable bias of cryo-EM SPA or is it real?

-Comparison with the mid-point of the transition measured on GLIC in liposome (Ref. 37) Actually this measurement could have been biased by the fact that the curvature of the lipids in liposome is completely different from the one in a planar membrane.

See line 100, line 488, line 596.

-Effect of the internal pH

The reference cited p. 19 (line 594-595) to argue that the pH has a minimal effect on the TMD in activating the channel is cited out of context and is not really appropriate here: what is shown in this article is that titrable residues in the TMD (glutamates) have little or no role in the transition, but long-range effects of the difference of pH in the inner cell and outer cell compartments cannot be ruled out. Actually, in van Renterghem et al., 2023

<https://physoc.onlinelibrary.wiley.com/doi/10.1113/JP283765>

it is shown that intracellular pH does modulate the activation transition.

-On the existence of a D-state

Line 483, the authors write:

“It is intriguing that GLIC has refrained from adopting a desensitized state after being present in an activating condition for an extended period of time in our current experimental setup.”

Also, line 589:

“The main limitation of this study is not being able to capture a desensitized state, although the protein is at low pH conditions for an extended period of time.”

This is indeed true: it is surprising that the D-state was not obtained. The present reviewer does not think that solving the structure of GLIC in a liposome could remediate this situation, as it will also lack a gradient of monovalent cation concentration across the membrane. This part of the Discussion should be revised.

Details

-p. 4 The pH₅₀ in liposome was “calculated” => “measured”?

-p. 6 line 181: “twisted clockwise”: by how much? See also p. 13 line 420

-p. 11 Ref 47 and its PDB file was in fact available in Sept. 6th?

-p. 15 “Although this region is narrower...” it is actually 2.71 Angstrom instead of 2.78 : perhaps it could be argued that small movements allowed by value of the B-factor could be sufficient to ensure ion passage?

-p. 15 line 469: Needs a Reference

-p. 15 line 478: “which is...” (typo)

-p. 15 line 481: “might have been ...” (typo)

-p. 16 line 501: “occupancy” (typo)

- p. 16 line 520-521: Ref. 49 is about propofol and bromoform (not desflurane)
- p. 17 line 538 "PDB is not available...": in fact it is available
- p. 28 The format of Reference 80 is wrong

Reviewer #3 (Remarks to the Author):

For the revision, the team collected new, larger datasets and reprocessed existing data, resulting in resolution improvements. In the case of low-pH conditions, the new datasets resulted in a putative activated open channel structure and two intermediates. These result in a strengthened study with the presentation of structures likely corresponding to states along the/an activation pathway. Figures are improved and especially those related to EM processing and map quality are excellent. However, this study still feels far from ready for publication, related to the writing, and for this particular journal, clear conceptual advance. The explicit premise of the study is that GLIC is a more tractable receptor study and what we learn from it will teach us about how the eukaryotic receptors gate. However, what is learned here from GLIC is not compared/contrasted with the eukaryotic receptors, where in fact several have been subjected already, successfully, to studies of gating mechanisms by cryo-EM. I list more specific issues below, including disconnects between what is expected from physiology and what is observed in the structures.

1. The overall presentation remains unpolished. Figures are not prepared according to a journal style (are they really all full-page figures? Fig. 5 could be dramatically condensed), which, to a degree, makes the reviewers guess at what the final versions might look like. The writing remains rough and imprecise. Lines 75-76 as one example of a sweeping statement that is unreferenced and inaccurate: "Irrespective of the ion selection properties of pLGICs, the fundamental mechanisms of activation and ion conduction are analogous." In fact, there is a surprising amount of diversity in the family in gating mechanism. GLIC is gated by protons, which seem to not bind in the same orthosteric site as, for example, 5-HT (see Hu...Delarue paper referenced in the MS). There are two gating mechanisms from structures proposed for 5-HT3, and they involve TMD twists in opposite directions. Eukaryotic cation-selective channels with a partially ordered ICD appear to gate differently from prokaryotic cation channels and eukaryotic anion channels. Recent work, both structural and functional, suggests the possibility of asymmetry in gating in some heteromeric pLGICs and even a homomeric pLGIC. It's fair to say it was generally expected that the gating mechanisms would be conserved across the family, but there have been many surprises. Please be more precise about what is intended when referring to the gating mechanism. Carefully read each sentence for accuracy and precision in meaning. Another example: "solving" structures is relevant to solving the electron density equation for the phases in crystallography, but is not a relevant term for cryo-EM 3D reconstructions.

2. Line 98-99, premise of the study. There are now several eukaryotic pLGICs where gating mechanisms are being probed directly; how does using GLIC provide advantages? Could studying GLIC shed light on

proton sensing? Could studying GLIC shed light on evolution of the channel family? There are special and important aspects of GLIC, but using GLIC as a surrogate to understand eukaryotic pLGIC gating is challenging to justify- and, what is learned from GLIC in the current study is not applied to understanding eukaryotic pLGICs.

3. That the apo structures are different, between detergent and nanodisc conditions, only in the ECDs, is surprising. Why might this be? This major finding from the first section ends as a perplexing observation. How do we conclude that one ECD is a better representative of a 'true' apo-resting state?

4. Movies that include structural morphs are needed to effectively illustrate the cascade of conformational changes- overall and region-by-region.

5. GLIC is known to desensitize at steady state at low pH. How is this to be reconciled with an absence of closed-desensitized conformations obtained in this study? Do the lipids used in cryo-EM support function in, for example, liposomes? Should different nanodiscs be tested? The Cheng group's preprint suggests MSP1E3D1 constrains conformational transitions more than spMSP1D1, for another bacterial pLGIC. <https://www.biorxiv.org/content/10.1101/2022.11.20.517256v1>

6. The section on lipid interactions (leading up to the discussion) is highly detailed and lacks mechanistic conclusions. These are nicely included in the discussion, when references are available. Condensing the lipids results section so it feels less like long list would increase readability.

7. Comparisons with a reference eukaryotic pLGIC could strengthen interpretation of what an apo-resting state ought to look like (lines 428-430). The twisting could be compared with 5-HT3 to help understand which proposed gating mechanism of 5-HT3 is most similar to GLIC, or proposed mechanisms in GlyR where again many structures are available, or the $\alpha 7$ nicotinic receptor.

8. Please limit the use of the word unprecedented. Or, clarify what is meant. Is a given unprecedented result a surprising one, which then needs to be validated? Or one that is finally consistent with expectations from other systems? (Maybe still needs to be validated, at least through comparisons).

9. The statement about "open probability of 100%" is misleading. I think what is meant is that only that conformation is able to be seen in the EM data processing, but "open probability" suggests a functional experiment. The open probability of GLIC in a membrane at low pH is presumably low, at equilibrium (considering long shut times as well as clusters of openings), due to desensitization. That this is not observed in the structure suggests that the reconstitution conditions are imperfect- worth acknowledging. I see this acknowledgement now in the "Limitations" section. Thanks for including. Please be precise about what context 100% open probability refers to.

10. Fig 7 cartoon M2 helices- it looks here like all structures are more closed at the cytosolic end, but structures suggest that most structures are more closed at the extracellular end.

11. SF2b suggests large gaps in ECD subunit interfaces. Is this only seen in GLIC with asolectin?

12. SF6 suggests multiple pentamers in a nanodisc- surprising? Could this affect conformational states observed?

13. A gating mechanism should, ideally, start with agonist binding, but this step appears to be missing from the presentation in the manuscript. Do the structures lend insight into how the key proton sensor identified in PMID 30541892 may trigger conformational transitions?

Reviewer #1 (Remarks to the Author):

The manuscript by Bharambe and colleagues has significantly improved, revealing more structural intermediates of the activation cycle of GLIC at an enhanced resolution. The authors have now demonstrated a wider pore in one of the structures at pH 4 and 2.5. Consequently, they could align the structures in an apparent activation sequence. The manuscript is also much more readable and has better references to the literature.

However, there are some serious concerns that need to be addressed before the manuscript can be published.

Major Concerns:

1. My primary concern is the assignment of the structures to their presumed functional roles. GLIC may desensitize even without cholesterol, and the structures of the desensitized receptors will likely look more similar to the open state than to the closed ones and will be unable to pass ions. In the current manuscript, the justification for assigning the structure to intermediate and pre-open states is unclear and requires further work.

In our cryo-EM experiments, GLIC adopts multiple conformational states at pH 4.0, which are the C1, C2, and O states. The pore of the O state reflects an open state conformation, which is supported by MD simulations. In these circumstances, one could assume that the C1 and C2 states could represent either pre-open or post-open (desensitized) states. However, at pH 2.5, the population of the O state dominated, and the other two states were not observed. Therefore, we believe that the conformation of O state represents a functional state that is ahead of the other two states in the activation pathway. Additionally, accordingly to the current understanding of GLIC and other eukaryotic pLGIC structures, the desensitized state has a constricted pore below 9' particularly near the -2' region (PMID: 25891813, PMID: 32719334 and PMID: 29484660), which is not observed in both C1 and C2 states. During the revision period, we performed additional MD simulations of the C1 and C2 states, which show dehydration at 19' in both structures, which is similar to our apo state (Supplementary fig. 14). Therefore, absence of pore constriction below 19' and de-wetted pore at the 19' observed during simulations suggest that C1 and C2 represent states that precede the open conformation. The analysis and explanations have now been included in the discussion section of the manuscript.

In line 500: "If C1 and/or C2 are desensitized state, then these states would be predominantly observed at pH 2.5. However, all the intact particles on the cryo-EM grid adopt a conformation very similar to the GLIClipids-pH4.00 (Supplementary fig. 11). This observation clearly demonstrates that both C1 and C2 precede the O state..."

2. On line 215, the authors built water molecules into the EM maps at a resolution of 2.5-2.7 Å. I have serious concerns about whether this can be reliably done. In fact, publications suggest that reliable building of water molecules can be done at a resolution better than 2 Å (PMID: 34473085). Thorough validation would be required to make placement of water in such maps close to the symmetry axis (like in the pore in Figure 4g) reliable. I did not find such validation in the manuscript text.

Thank you for your valuable suggestions. To clarify, we have not built water molecules in both pH 7.5 maps, which have a resolution range of 2.9–3.4 Å. The nominal resolution of GLIC_{lipids}-pH4.0_O and GLIC_{lipids}-pH2.5_O is ~2.7 Å. However, we have recalculated the local resolution in the range of 2-4 Å using Resmap. This reveals that the resolutions in most of the regions are at least 2 Å (Supplementary figure 9b and 12b). Therefore, it justifies building waters in these maps. The important point to note here is that except pentagons 1-4 (observed in open state only), all other water molecules were identified previously. The water pentagons 1-4 are visible mainly because of the absence of detergent molecules in that region, and they are loosely packed. Based on previous MD simulations, the water molecules in this region were stabilized by complex polar interactions (PMID: 23403925). In our open structure, each water molecule in pentagons 2–4 is primarily stabilized by interacting with adjacent layer (Fig 4h). The water molecules in pentagon 1 apparently do not involve any polar interactions. However, several densities (water or ions) are present in the vicinity, which might stabilize the water molecules in pentagon 1 (Fig 4h: inside box). Waters in pentagons 5 and 6 are stabilized by interacting with the S6' and T2' located in M2 (Fig 4h). We have mentioned the water molecules in more detail in the results section of the revised manuscript.

Line 333: “The water molecules in pentagons 2-4 are stabilized by polar interactions with water molecules present in the adjacent layer”

3. The fact that MD simulations showed good hydration of the pore (line 328) by water molecules does not necessarily mean that it is ion-permeable. The authors base their assignment of GLIC_{lipids}-331 pH4.0_O as an open state; however, more evidence is required that this state allows ion passage.

We appreciate your concern. We have performed MD simulations on the pH4 open state, which shows Na⁺ ion permeation events. This clearly indicates that the open state is ion permeable. We have included the new data in the main text and figures (Fig. 5e-f).

Line 356: “Several Na⁺ ion permeation events were observed when the same experiment was performed on GLIC_{lipids}-pH4.0_O (Fig. 5f).”

4. The authors describe the interactions of lipids with the channel; however, they do not seem to draw any conclusions from these interactions. A series of point mutations on GLIC could answer these questions. Some mutations have been reported in the literature and are discussed in the text; others could be performed based on new structural findings.

Below is a list of residues that interact with lipids and their known functional roles in GLIC.

Name and position of residue	Location of the residue	Interacting lipid(s)	Known functional role	Source of study
R118	β6-β7 (pro-loop)	Lipid 2	R118A mutant shows loss of function.	PMID: 26318456

Y194	pre-M1	Lipid 2 and lipid 9	The pH ₅₀ of Y194A mutant is similar to WT	PMID: 24367074
F195	pre-M1	Lipid 9	F195A mutation shows marked loss of function.	PMID: 24367074
N200	M1	Lipid 1	Not available. However it's counterpart in $\alpha 1$ GlyR, Q226, shows reduced agonist response when mutated to Alanine.	PMID: 20023641
P250	M2-M linker	Lipid 9	P250A same to WT	PMID: 24167270
M252	M2-M3 linker	Lipid 1a	M252A mutant has similar pH ₅₀	PMID: 24167270
T253	M2-M3 linker	Lipid 1b	T253A mutant shows no conductivity	PMCID: PMC3832033
Y278	M3	Lipid 5	Not available	Not applicable
R287	M4	Lipid 3, lipid 4	R287A mutant shows reduced pH ₅₀ which is statistically significant.	PMID: 26318456
R293	M4	Lipid 5	R293A mutant shows reduced pH ₅₀ which is statistically significant.	PMID: 26318456

As shown in the table above, all the interacting residues were previously mutated, except N200 and Y278 (highlighted in green). The E243 in GLIC and the equivalent position of N200 in the glycine receptor play a critical role in stabilizing the open state as confirmed by the mutational studies (PMID: 30541892, PMID: 29281623, PMID: 22580559 and PMID: 24405574) and this knowledge can be translated to GLIC and we have mentioned this in the text.

Line 550: "The absence of Lipid 1-N200 interaction and outward movement of M2 helix facilitate.."

Y278 is located in the lower leaflet. Our observation based on the conserved locations of lipids present in the lower leaflet indicates their limited role on the state specific interactions. This hypothesis is also supported by the recent work of Erik Lindahl's group (<https://doi.org/10.7554/eLife.86016.1>). However, in almost all structures, Lipid 5 interacts with Y278, which was predicted to be involved in the interaction with neurosteroids and potentiate the GLIC activation (PMID: 29301936), indicating its possible role in channel function. To the best of our knowledge, Y278 has not been mutated to investigate its modulatory effect on GLIC. Unfortunately, mutation of this residue to investigate its role on the channel function is not feasible in our lab at the moment.

In our revised manuscript, we have now condensed the lipid part in the results section and rewrote the discussion section for better conclusion. We summarized that Lipids 1, 2, and 9 interact with proteins in a state specific manner. Especially, Lipids 2 and 9 are the most critical lipids which might be responsible for channel gating.

5. For the next round of reviews, it would be important to provide validation reports from depositions of structures to publicly accessible databases.

All maps and models are deposited in publicly accessible databases. The validation reports are shared with the editor and are available for review.

Minor Concerns:

1. The statement that reported apo-structures are in previously unseen conformations (line 194) is obvious and somewhat redundant as it is described in the preceding paragraph. Should this be removed or discussed in more detail?

We have rephrased it to “Taken together, our apo state is captured in a distinct conformation that shows an expanded and clockwise twisted ECD compared to the existing apo states.”

2. The authors report a second digit at resolutions of 2.7 Å. How relevant is this?

We have now changed all reported resolutions to single digit.

3. I am unsure what the authors mean by “holistic glimpse” (line 580).

The sentence is now rephrased to “In summary, our structures provide a better understanding of proton induced conformational transitions, the gating mechanism, and state-dependent protein-lipid interactions in GLIC at the molecular level.”

Reviewer #2 (Remarks to the Author):

Comments on the revised manuscript of Basak et al. Nature Commun. Oct 2023

Using cryo-EM the authors recollected more data of GLIC reconstituted in a nanopore and obtained a continuum of structures of very good quality between the closed form and the open form, in a lipidic environment, by varying the pH.

They describe very interesting new experimental structures, especially for the channel of the open form with very nice density for several water pentagons in the pore as shown in Fig. 13h.

I still have important concerns to share about the present state of the manuscript.

In particular a number of important papers relevant to the present manuscript are not discussed properly.

-Position of lipids

The paper in eLife by Lindahl and coll. (Ref. 47) should be discussed more fully, as it reports also the position of the lipids for at least one form of GLIC, and this is one of the main points of the manuscript. Their PDB structure 8ATG is in fact publicly available since Sept. 6th.

The authors should use it and compare the positions of lipids observed here to those they observe. See Fig. 14 and 15, which should be changed.

p. 11 line 347 “However, a comparison of the lipids was not feasible due to unavailability of the model, at the time of manuscript preparation...”

p. 17 line 538 “(PDB is not available)”: In fact, it is available.

Their full MD trajectories are also actually available since April 25th.

We have now discussed this paper in more detail. The group has combined all the cryo-EM data that were previously collected at different pH conditions (PMID: 34210687) and re-processed together. They identified very few particles (~16,500 compared to previously used ~651,000 particles for three pH conditions) that yielded relatively high-resolution map with five lipid densities in each subunit (PDB ID: 8ATG). Therefore, it is difficult to correlate the functional state of this structure with a particular pH condition. Nevertheless, it is incredible that several lipids have been observed in the TMD, many of which were not shown previously. The overall structure (8ATG) is similar to the previously determined structures (6ZGD) with a backbone RMSD of ~0.45 Å. In the revised manuscript, we have compared the lipids with our structures. The locations of lipids 2, 3, and 4 are similar to 8ATG. The head group of Lipid 1 is located deeper toward the membrane in our structure compared with 8ATG. The density of Lipid 2 acyl chain is resolved longer in our pH4 states compared to 8ATG. Location of Lipid 5 differs in 8ATG. This lipid is away from the protein surface in 8ATG and apparently does not interact with the protein. Lipids 6, 7, and 9 are extra lipids that we found in our structures. Lipid 9 is located in the upper leaflet and is observed only in the C2 and O states. Lindahl’s group found another lipid in the open state using MD simulations. Interestingly, our Lipids 9 is positioned very close to the predicted lipid (Supplementary fig. 15l). Therefore, in our lipid 9 highly correlates with the simulated lipid, and this is now also compared in the manuscript. We have also updated the sentences in which we stated previously that the data are not available. We have updated and included several sentences that focus on discussing lipids in 8ATG. Below are a few representative sentences that have been modified or added in the revised manuscript.

Line 522: “Lipids 1-5 are also observed in the recent cryo-EM structure of GLIC where the locations of Lipids 1 and 5 are marginally different (Supplementary fig. 15j-k).”

Line 558: “In a recent study, MD simulations predicted the presence of an additional lipid in the open state of GLIC at the upper leaflet, which wedges between two subunits and the predicted location is close to our Lipid 9 (Supplementary fig. 15l).”

In fact, the statement line 580

‘In summary, our structures provide the first holistic glimpse of the gating mechanism and state-dependent protein-lipid interactions in GLIC’

should be toned down as there has been a number of studies on the gating mechanism of GLIC (Ref 9), as well as on “state-dependent protein-lipid interactions (Ref 47, Figure 4).

Thank you for this suggestion. The sentence has now been rephrased to “In summary, our structures provide a better understanding of proton induced conformational transitions, the gating mechanism, and state-dependent protein-lipid interactions in GLIC at the molecular level.”

-Diameter of the pore

Figure 5: The reported diameter of the water molecule (dotted line) seems to be wrong: it should be 2.78-2.8 Angstrom, not 2.5 A. See also p. 33 the Legend of the Figure 5.

We apologize for the error. We have now fixed the figure.

-Scenario for ion channel gating and ion permeation

How does the scenario described in Figure 7 and in the Discussion compare with the experimental work of Menny et al. 2017? See <https://pubmed.ncbi.nlm.nih.gov/28294942/>

Thank you for your suggestion. In our experiment, at pH 5.5, the channel undergoes counter-clockwise rotation and compaction, whereas no conformational changes are observed at the M2-M3 loop. In the paper, they found that there is a global compaction in ECD at pH 5, and the conformational changes are nearly completed very fast while the channel is closed. Therefore, they called this state pre-active intermediate state. They observed the outward movement of the M2-M3 loop. Because the extent of M2-M3 movement was not measured, their intermediate state might be similar to our the C2 state. We have mentioned this in the revised manuscript.

Line 599: "Recently, fluorescent quenching experiments revealed similar compaction of ECD and outward movement of the M2-M3 loop upon protonation while the channel is closed, which was referred to as the pre-active state."

How does it compare with T. Allen and coll. article on GLIC transition studied by the string method in MD simulations in 2017? See <https://pubmed.ncbi.nlm.nih.gov/28487483/>

Thank you for the suggestion. In the string method they have investigated the conformational changes from open to closed state using MD simulations. The key observation is that the high extent of ECD twist is required for channel opening. We observed exactly the same. The ECD of C2 state (pre-open) is significantly different than apo state. They observed global expansion in ECD and β -sandwich expansion reflected by breaking D32-R192 interaction (a part triad formed by D32-R192-D122 in open state (PMID 26943937, PMID 30541892)) in closed state. This interaction is important for channel function (PMID: 26943937). In our C2 state this triad is formed. In fact in our O state, the β 1- β 2 loop moves downwards and interact more strongly with R192 (Fig. 2j). The similar interaction is not observed in remaining of our closed structures. During MD simulations they found an open state which is wider than the existing open state. But we observe a subtle constriction in our open state compared to existing one. However, they observed the pore of closed state (apo) is less closed which is in line with our observation. They found several intermediate states. I_1 state has expanded ECD with semi-open (non-conducting) pore which we haven't observed. Other proposed intermediate states also seem to be different than ours. However, we have mentioned in the text that several other intermediate states could exist, whose structures are required to fully understand the conformational transitions upon activation.

Line 255: "In addition to notable inter-subunit interactions at the ECD, an intra-subunit triad is established...."

Line 306: “Even though the pore is apparently wider in our apo state than GLICdetergent-pH7.5C, which is consistent with previous study ...”.

Line 585: “Due to the complexity of gating mechanism, which involves several unresolved intermediate states, assignment of functional states in a sequence along the activation pathway is challenging.”

In particular, the transition described in this PNAS paper is not symmetrical, each monomer undergoing a transition in a progressive manner. In contrast, all structures presented here have C5 symmetry by construction. Is this an inevitable bias of cryo-EM SPA or is it real?

Thank you for raising these valuable points. During the string method, they observed the asynchronous movement of M2 during channel closure. However, when we processed our data in C1 symmetry, the asymmetric transitions were not observed in any of our data when we compared them with C5 symmetry reconstructed maps. For better understanding, instead of overlaying maps (C1 Vs C5), which is difficult to interpret from images, we built models using maps processed in C1 symmetry. Models for only pH4 maps are shown below. The new models (built from C1 symmetry maps) were compared with the submitted models (generated from C5 symmetry), and residue-by-residue RMSD was calculated (Reviewer figure 1). There is no alteration in the conformation, indicating that both C1 and C5 maps are identical. This figure is only for the reviewer’s reference and has not been incorporated in the main manuscript. If the asymmetric state channel is real but we don’t observe in our cryo-EM experiments, the reason might be that the populations of asymmetric states might be very low and difficult to identify during data processing.

-Comparison with the mid-point of the transition measured on GLIC in liposome (Ref. 37) Actually this measurement could have been biased by the fact that the curvature of the lipids in liposome is completely different from the one in a planar membrane.
See line 100, line 488, line 596.

The authors of the paper (Ref 37) prepared giant liposomes for functional studies. Therefore, the effect of curvature on the functional study would be minimal. However, our goal was not to justify pH_{50} in liposomes. Rather, this pH_{50} encouraged us to perform cryo-EM experiments with the expectation that it would stabilize the desensitized state below pH 2.9. However, lowering pH enhanced the gating equilibrium toward the activation pathway, which agrees with the aforesaid experiment (Ref 37), but ultimately stabilized all the intact particles into open states on our cryo-EM grid. Therefore, in the future, we will focus on stabilizing the desensitized state.

-Effect of the internal pH

The reference cited p. 19 (line 594-595) to argue that the pH has a minimal effect on the TMD in activating the channel is cited out of context and is not really appropriate here: what is shown in this article is that titratable residues in the TMD (glutamates) have little or no role in the transition, but long-range effects of the difference of pH in the inner cell and outer cell compartments cannot be ruled out. Actually, in van Renterghem et al., 2023 <https://physoc.onlinelibrary.wiley.com/doi/10.1113/JP283765> it is shown that intracellular pH does modulate the activation transition.

Sorry for the misunderstanding. Thank you for highlighting this important paper. We agree that the long-term allosteric effect of internal pH (pH_i) cannot be ruled out. Low pH_i has an inhibitory effect on channel gating. Although the mechanism of inhibitory effect is not understood, but this could be one reason why we see slow transition rate in the activation pathway. We observed only the open state, but not the D state, which could be due to the low pH_i . The authors also mentioned that even in liposomes (during electrophysiology experiments), there could be an influx of protons when the channel opens, which could lead to inactivation and, therefore, might be misunderstood as a desensitized state during functional study. We have mentioned the effect of intracellular pH on gating in limitation section of the revised manuscript.

Line 622: “Although a previous study showed that the TMD does not possess the principal component of proton activation³⁸, the inhibitory effect of long-term exposure to internal low pH (pH_i) has been demonstrated previously^{50,82}”

-On the existence of a D-state

Line 483, the authors write:

“It is intriguing that GLIC has refrained from adopting a desensitized state after being present in an activating condition for an extended period of time in our current experimental setup.”

Also, line 589:

“The main limitation of this study is not being able to capture a desensitized state, although the protein is at low pH conditions for an extended period of time.”

This is indeed true: it is surprising that the D-state was not obtained. The present reviewer does not think that solving the structure of GLIC in a liposome could remediate this situation, as it will also lack a gradient of monovalent cation concentration across the membrane. This part of the Discussion should be revised.

Thank you for this valuable suggestion. The low pH_i could also be the reason why the D-state is not populated. Renterghem et al. has also raised the question of defining the D-state in GLIC. The decay in current after activation could be due to inactivation at low pH_i and misunderstood as the D state. However, the inactivated state is not observed in our study at a very low pH (2.5). They hypothesized that “a low- pH_o induced, dropping- pH_i (rising- H^+)-mediated channel inactivation appeared useful to a cell’s survival, and that evolution developed the proper agonist-compound-induced desensitization”. Another hypothesis they made is that the D state would be recruited after long exposure to positive allosteric modulators (PAMs such as FUMAR). This hypothesis could be explored in the future using our nanodisc-reconstituted GLIC. The monovalent cation gradient can be generated by washing the liposome-reconstituted GLIC with a higher salt concentration. Although proton influx cannot be avoided, the rate of pH drop inside liposomes would not be as intense as exposing TMD to low pH in non-liposome preparation. PAM can be tested on liposome-reconstituted GLIC. We have revised this section and proposed a strategy to stabilize the D-state.

Line 628: “However, upon activation, GLIC could also permeate protons and decrease”

Details

-p. 4 The pH_50 in liposome was “calculated” => “measured”?

Changed.

-p. 6 line 181: “twisted clockwise”: by how much? See also p. 13 line 420

The extent of clockwise twist varies in various regions. Therefore, we have mentioned the angle of twisting in certain regions that show large changes. We have now included more regions, measured the angle of rotation, and discussed this throughout the manuscript.

Line 188: “Moreover, Loop A moves upward in our apo state with a maximum Ca distance of $\sim 3.1 \text{ \AA}$ and also twisted $\sim 8^\circ$ clockwise.”

Line 229: “A gradual but noticeable counter-clockwise rotation is prominent in the $\beta 1$ - $\beta 2$ loop at lower pH, with the maximum rotation of $\sim 16^\circ$ observed at O state compared to apo state.”

Line 232: “Loop C gradually moves inward toward the central axis and adopts a closed conformation at all low pH states, with the minimum and maximum (~11°) rotation...”

-p. 11 Ref 47 and its PDB file was in fact available in Sept. 6th?

This sentence has been removed, and the new lipids are now compared and discussed in the revised manuscript.

-p. 15 “Although this region is narrower...” it is actually 2.71 Angstrom instead of 2.78 : perhaps it could be argued that small movements allowed by value of the B-factor could be sufficient to ensure ion passage?

Yes, it is possible that small movements of the sidechain could also allow the ions to pass through. We acknowledged this in the main text as follows.

Line 468: “A small fluctuation of the sidechain location of T2’ away from the pore axis could also allow the ion to pass through.”

-p. 15 line 469: Needs a Reference

Reference included.

-p. 15 line 478: “which is...” (typo)

Fixed.

-p. 15 line 481: “might have been ...” (typo)

Fixed.

-p. 16 line 501: “occupancy” (typo)

Fixed.

-p. 16 line 520-521: Ref. 49 is about propofol and bromoform (not desflurane)

Yes, Ref 49 is about propofol and bromoform and Ref 40 is about propofol and desflurane.

-p. 17 line 538 “PDB is not available...”: in fact it is available

Fixed.

-p. 28 The format of Reference 80 is wrong

Fixed.

Reviewer #3 (Remarks to the Author):

For the revision, the team collected new, larger datasets and reprocessed existing data, resulting in resolution improvements. In the case of low-pH conditions, the new datasets resulted in a putative activated open channel structure and two intermediates. These result in a strengthened study with the presentation of structures likely corresponding to states along the/an activation pathway. Figures are improved and especially those related to EM processing and map quality are excellent. However, this study still feels far from ready for publication, related to the writing, and for this particular journal, clear conceptual advance. The explicit premise of the study is that GLIC is a more tractable receptor study and what we learn from it will teach us about how the eukaryotic receptors gate. However, what is learned here from GLIC is not compared/contrasted with the eukaryotic receptors, where in fact several have been subjected already, successfully, to studies of gating mechanisms by cryo-EM. I list more specific issues below, including disconnects between what is expected from physiology and what is observed in the structures.

Thank you for your valuable comments and recognizing the improvement of our work. Below please find point-by-point explanation of your concerns.

1. The overall presentation remains unpolished. Figures are not prepared according to a journal style (are they really all full-page figures? Fig. 5 could be dramatically condensed), which, to a degree, makes the reviewers guess at what the final versions might look like. The writing remains rough and imprecise. Lines 75-76 as one example of a sweeping statement that is unreferenced and inaccurate: “Irrespective of the ion selection properties of pLGICs, the fundamental mechanisms of activation and ion conduction are analogous.” In fact, there is a surprising amount of diversity in the family in gating mechanism. GLIC is gated by protons, which seem to not bind in the same orthosteric site as, for example, 5-HT (see Hu...Delarue paper referenced in the MS). There are two gating mechanisms from structures proposed for 5-HT₃, and they involve TMD twists in opposite directions. Eukaryotic cation-selective channels with a partially ordered ICD appear to gate differently from prokaryotic cation channels and eukaryotic anion channels. Recent work, both structural and functional, suggests the possibility of asymmetry in gating in some heteromeric pLGICs and even a homomeric pLGIC. It's fair to say it was generally expected that the gating mechanisms would be conserved across the family, but there have been many surprises. Please be more precise about what is intended when referring to the gating mechanism. Carefully read each sentence for accuracy and precision in meaning. Another example: “solving” structures is relevant to solving the electron density equation for the phases in crystallography, but is not a relevant term for cryo-EM 3D reconstructions.

Thank you for your concerns and valuable comments regarding the figures. In most cases, the width and image quality were maintained according to the Nature Communications guidelines. If we make the width larger (which we is also found in few other Nature Communications papers), it will not look like a full page. Therefore, we are happy to work with the Nature Communications team to fix and fulfil the criteria according to their guidelines. Figure 5 has now been modified. We agree that there are many specific differences between prokaryotic and eukaryotic pLGICs. Even among the eukaryotic pLGICs, there are differences. However, there is a consensus that upon ligand binding in the ECD (proton in case of GLIC or neurotransmitters in other pLGICs), the ECD undergoes counter-clockwise rotation and compaction (PMID 36914669, PMID 33735609, PMID 30401837, PMID 26344198), which allosterically propagates to the TMD to open the pore lined by M2 helices. The direction of TMD rotation may not be conserved (based on the 5-HT₃R), but the channel opens at the gate located at 9' position and subsequently conducts ions. Although there are some differences in specific

conformational changes, the fundamental mechanisms of agonist-induced conformational changes in ECD and channel pore opening are similar. However, in the revised manuscript, we have carefully rephrased the sentences to improve precision. We have removed the sentence “Irrespective of the ion selection properties of pLGICs, the fundamental mechanisms of activation and ion conduction are analogous”. We have now removed the term “solve” wherever it is not appropriate.

We have rephrased several sentences to improve the manuscript. Some of them are highlighted below.

Line 129: “The structure of GLIC at pH 3.0 reconstructed from the minor populations of protein particles shows...”.

Line 133: “On a cautionary note, the poor resolutions (~5 Å) of the maps reconstructed from the minor populations protein particles might have yielded.....”.

Line 165:” We obtained two cryo-EM structures of GLIC reconstituted in soybean polar lipid extract.....”.

2. Line 98-99, premise of the study. There are now several eukaryotic pLGICs where gating mechanisms are being probed directly; how does using GLIC provide advantages? Could studying GLIC shed light on proton sensing? Could studying GLIC shed light on evolution of the channel family? There are special and important aspects of GLIC, but using GLIC as a surrogate to understand eukaryotic pLGIC gating is challenging to justify- and, what is learned from GLIC in the current study is not applied to understanding eukaryotic pLGICs.

Recently, Pierre-Jean Corringer group published a paper on GLIC (PMID: 37026398) and stated that “GLIC and ELIC appear as excellent models for mammalian pLGICs because their main structural, biophysical, and pharmacological properties are very well conserved”. There are also several studies that have mentioned the same (PMID 27151638, PMID 32627739). However, we completely agree with the reviewer that there are differences between prokaryotic and eukaryotic pLGICs. Therefore, we have toned down the text in the revised manuscript. The aim of this study is to better understand the gating mechanism of GLIC in a lipidic environment. We have made this clear throughout the main text. The key findings of our study are as follows. We captured an apo state whose conformation is different from that of the existing detergent solubilized apo state. Thus far, intermediate states (like locally closed structures) have been stabilized by mutations or by cross-links (PMID: 22580559). By capturing intermediate states of wild-type GLIC, we have a better understanding of the structural transitions of GLIC upon activation in the lipidic environment. We also obtained an open state that does not have detergent trapped in the pore. Several lipid densities have been resolved now and provide new insights into cross-talk between protein and lipids. Therefore, our study provides a comprehensive overview of proton-induced channel opening in the lipid environment. However, the GLIC also has a special interest. In the aforesaid paper (PMID: 37026398), the authors also discussed in detail about the presence of orthosteric/orthotopic binding site where the positive allosteric modulators di-carboxylate compounds bind. Although GLIC is activated by protons and the proton binding sites are different from the orthosteric/orthotopic sites, they may have evolved to be the primary binding sites for neurotransmitters and agonists in eukaryotic ligand gated ion channels. GLIC also possesses vestibular sites that can also be modulated by PAM (PMID: 37026398). This vestibular site is also present in eukaryotic pLGICs and could be targeted for drug development.

3. That the apo structures are different, between detergent and nanodisc conditions, only in the ECDs, is surprising. Why might this be? This major finding from the first section ends as a perplexing observation. How do we conclude that one ECD is a better representative of a ‘true’ apo-resting state?

*According to previous studies, the activation of GLIC and other pLGICs upon agonist binding starts with counter-clockwise rotation and compaction in ECD (PMID 36914669, PMID 33735609, PMID 30401837, PMID 26344198, PMID 24367074, PMID 34224751, PMID 33735609, PMID 30602790). These conformational changes propagate to the TMD and cause pore opening and subsequent ion conduction. Therefore, the state that is closer to the resting state should have an expanded ECD. Compared with the detergent-solubilized apo state, our nanodisc-reconstituted GLIC in the apo state has an expanded ECD that is twisted clockwise. Therefore, we believe that our apo state is closer to the resting state. We **refrained** from claiming that it is a **true** resting state throughout the main text and have called it a resting-like state. In Figure 7 and other places, wherever we unintentionally mentioned resting state previously, is now rephrased to resting-like state.*

4. Movies that include structural morphs are needed to effectively illustrate the cascade of conformational changes- overall and region-by-region.

We have included movies that show global changes and cascade of conformational changes in important regions.

5. GLIC is known to desensitize at steady state at low pH. How is this to be reconciled with an absence of closed-desensitized conformations obtained in this study? Do the lipids used in cryo-EM support function in, for example, liposomes? Should different nanodiscs be tested? The Cheng group’s preprint suggests MSP1E3D1 constrains conformational transitions more than spMSP1D1, for another bacterial pLGIC. <https://www.biorxiv.org/content/10.1101/2022.11.20.517256v1>

nAChR and ELIC are highly sensitive to lipid composition (PMID 31724949, PMID 6746645, PMID 7512384). These channels adopt uncoupled conformational states when reconstituted in PC only or other specific lipids (PMID: 19357079 and PMID: 25519904). GLIC is less sensitive to lipid composition. We have captured the apo state and several intermediate states that show concerted conformational changes (rotation and compaction in ECD) along the activation pathway. The MD simulation also revealed the elevated hydration in the pore at lower pH states (pH4.0 and 2.5). Now, we have performed MD simulation and showed that the open state is permeable to Na⁺ ions. Therefore, our open state captured in absence of detergent at the pore better represents one of the open conductive conformations. The number of particles that adopted an open state increased with lowering the pH to 2.5. Therefore, the GLIC structures in the presence of our lipid composition correlate with a previous functional study (PMID 22474322). However, why the channel does not undergo the D state remains an intriguing question. Stabilization of channels in the open state instead of the D state is not uncommon. For instance, the cryo-EM structures of 5-HT_{3A}R in both detergent and lipidic environments in the presence of 5-HT were determined by several groups captured in open state not in D state (PMID 30401837, PMID 30401839, PMID 33594077). In a recent paper, Pierre-Jean Corringer group discussed the D state in GLIC by questioning whether the proper D state is present in GLIC or not and provided insight on how the D state could be stabilized using positive allosteric modulators such as fumarate (PMID: 37026398). Therefore, this strategy could be used in future to stabilize the desired D state.

Regarding the preprint from Cheng group, they found that the diameter of the nanodisc has a role in modulating the stability of a particular conformational state. They found that the diameter of the nanodisc produced by MSPE3D1 is ~9.3 nm. The larger circularized nanodisc (spMSP1D1) makes bigger nanodisc with diameter of ~10.8 nm. They could capture the desensitized state of ELIC only by using spMSP1D1 but not by the shorter one. Therefore, they have concluded that larger nanodisc should be the best choice for stabilizing functional channel. We calculated the diameter of our nanodisc using the same strategy used by them and found the diameter of nanodisc of around 10.7 nm (Reviewer figure 2), which is similar to the nanodisc formed by spMSP1D1. Therefore, we believe that it is unlikely that our nanodisc has an effect on the functional state of GLIC.

6. The section on lipid interactions (leading up to the discussion) is highly detailed and lacks mechanistic conclusions. These are nicely included in the discussion, when references are available. Condensing the lipids results section so it feels less like long list would increase readability.

Thank you for the suggestion. we have now condensed it in result section to increase readability. Our hypothesis is that more intermediate states are present along the activation pathway and that the location of lipids varies, especially in the upper leaflet, particularly Lipids 1, 2 and 9. We believe that all intermediate states have not been fully captured in our cryo-EM experiments. However, we have attempted to provide a mechanistic overview based our structural data, which is included in the revised manuscript.

7. Comparisons with a reference eukaryotic pLGIC could strengthen interpretation of what an apo-resting state ought to look like (lines 428-430). The twisting could be compared with 5-HT3 to help understand which proposed gating mechanism of 5-HT3 is most similar to GLIC, or proposed mechanisms in GlyR where again many structures are available, or the $\alpha 7$ nicotinic receptor.

Thank you for your valuable suggestion. We have now explicitly mentioned that clockwise rotation and expansion of ECD are widely adopted features of resting-like state in the pLGICs family, with a list of references (PMID: 34224751, PMID: 30401837, PMID: 33735609, PMID: 36914669, PMID: 26344198, and PMID: 30602790) where structures of eukaryotic pLGICs show more expanded and clockwise rotated in ECD.

In Line 432: "Notably, the clockwise rotation and expansion of ECD in the resting-like state in relation to agonized states have been prevalently observed in the pLGICs family based on the high resolution structures."

8. Please limit the use of the word unprecedented. Or, clarify what is meant. Is a given unprecedented result a surprising one, which then needs to be validated? Or one that is finally consistent with expectations from other systems? (Maybe still needs to be validated, at least through comparisons).

We have now carefully modified these sentences and reduced the word "unprecedented."

9. The statement about "open probability of 100%" is misleading. I think what is meant is that only that conformation is able to be seen in the EM data processing, but "open probability" suggests a functional experiment. The open probability of GLIC in a membrane at low pH is presumably low, at equilibrium (considering long shut times as well as clusters of openings), due to desensitization. That this is not observed in the structure suggests that the reconstitution conditions are imperfect- worth acknowledging. I see this acknowledgement now in the "Limitations" section. Thanks for including. Please be precise about what context 100% open probability refers to.

Thank you for this suggestion. We have now removed the term "open probability" wherever necessary and replaced it with "open state".

10. Fig 7 cartoon M2 helices- it looks here like all structures are more closed at the cytosolic end, but structures suggest that most structures are more closed at the extracellular end.

We have now fixed this.

11. SF2b suggests large gaps in ECD subunit interfaces. Is this only seen in GLIC with asolectin?

The represented orientation created confusion. It is now fixed.

12. SF6 suggests multiple pentamers in a nanodisc- surprising? Could this affect conformational states observed?

We have carefully investigated all 3D classes during data processing. We have processed data in both C5 and C1 symmetry (C1 symmetry map has been not shown in the manuscript but available upon request). However, we identified only these three maps that showed different conformations and therefore, processed separately. Identifying multiple conformations from the same sample is not uncommon in cryo-EM experiments. Further classification of these particles did not lead to any new conformations. Therefore, it is unlikely that multiple pentamers would affect the observed conformational states.

13. A gating mechanism should, ideally, start with agonist binding, but this step appears to be missing from the presentation in the manuscript. Do the structures lend insight into how the key proton sensor identified in PMID 30541892 may trigger conformational transitions?

We have now mentioned the key proton sensors responsible for channel activation. See page 8 line 255 “In addition to notable inter-subunit interactions at the ECD, an intra-subunit triad is established due to a subtle counter-clockwise rotation and downward movement of $\beta 1$ - $\beta 2$ loop toward to TMD...”. These residues have been studied extensively, as verified in the mentioned paper (PMID 30541892). In that paper, the authors mentioned that “For pH-gating, however, there is an additional difficulty in assigning the protonation state of all Asp, Glu, and His residues in the two end states. Presently, it is not known with certainty which of these residues are protonated concomitantly with the conformational transition”. In our study the resolution of the maps does not allow to build proton on the model. Therefore, it is difficult to determine the state of protonation of those sensors. However, transitional states of WT GLIC along the activation pathway is identified in our experiment, which has previously seen in only GLIC mutants or by cross-linking. D32-R192 interaction has been shown to play an important role in signal transduction at the ECD-TMD interface and critical for channel function (PMID: 26943937). In our C2 and O states counter-clockwise rotation downwards movement of $\beta 1$ - $\beta 2$ loop induces the polar interaction, which involves D32 located in $\beta 1$ - $\beta 2$ loop close to the key proton sensor E35 involves in polar interaction and R192 located in pre-M1. This is the part of the triad that is important for channel function (PMID: 26943937). The similar interaction is not observed in the rest of our closed structures. Therefore, cross-talk between $\beta 1$ - $\beta 2$ loop and pre-M1 is important for channel opening. This has now been mentioned in the main text.

Line 99: “The channel is activated at low pH and the residues responsible proton sensing are located in ECD but away from the canonical orthosteric binding site.”

Line 103: “Although the ICD is intrinsically missing in GLIC, upon activation by protons, the channel undergoes conformational changes reminiscent of eukaryotic orthologs.”

Line 255: “In addition to notable inter-subunit interactions at the ECD, an intra-subunit triad is established....”

REVIEWER COMMENTS

Reviewer #1 (Remarks to the Author):

The updated version of the manuscript by Bharambe and colleagues has significantly improved and I only have some minor comments below.

1. The assignment of the ph2.5 structure to the pre-open state is still a hypothesis/author's consideration. I don't think that in this case "clearly demonstrates..." (line 500) should be used, I suggest a less uncompromising verb.
2. The figure 5A,B has the pore profiles horizontal, typically they are oriented vertically to match the structures.

Reviewer #2 (Remarks to the Author):

I am mostly satisfied by the answers of the authors.

However, I do not agree with the title, as it stands: The experiments described do not "reveal" a new mechanism for gating, they refine what has already been described, only it is now in an environment containing lipids.

Also, the authors should be cautious on the fact that other authors just found out that nanodisks can be misleading, and structural results on the same kind of bacterial pLGIC (ELIC) can vary depending on the size of the nanodisks:

<https://pubmed.ncbi.nlm.nih.gov/38167383/>

The authors should discuss this in their Discussion.

I would suggest instead a title taken from their Introduction

Cryo-EM structures of nanodisc-reconstituted prokaryotic ligand-gated ion channel GLIC provide insights on the gating mechanism in a lipid environment

Reviewer #3 (Remarks to the Author):

For the second revision, the team performed new analyses and revised the text, adjusted the figures, and added supplementary structural movies. I appreciate the thoughtful responses and edits from my and other reviewers' comments. Two rounds of review have now passed with major concerns from all three reviewers at each step. While the manuscript continues to improve, and the structural data are interesting, I still do not support publication of the current version. A major outstanding issue for me is that the writing still needs improvement- condense, remove non-essential qualifying terms, limit acronyms and jargon, increase precision and accuracy, generally be less informal. The title and abstract are examples of being too long, too detailed, without conveying a succinct and crystal clear conceptual advance appropriate for a broadly-read journal. The issues start at the very beginning. The first few lines of the introduction could be loosely interpreted to suggest that pLGICs in bacteria, and GLIC specifically, mediate fast chemical transmission between neurons, which they do not. I am concerned that as written the text will confuse readers or lead them to an incorrect understanding.

I also remain unconvinced about state assignment, as reviewer 1 mentioned. These channels desensitize; that state should be the most stable at low pH. I understand there are often disconnects we must live with between structural biology and physiology, but assigning the low pH 'closed' states to intermediates on the way to activation rather than to the more stable desensitized state(s) does not make intuitive sense to me.

Reviewer #1 (Remarks to the Author):

The updated version of the manuscript by Bharambe and colleagues has significantly improved and I only have some minor comments below.

Thank you for your valuable comments which improved the manuscript significantly.

1. The assignment of the ph2.5 structure to the pre-open state is still a hypothesis/author's consideration. I don't think that in this case "clearly demonstrates..." (line 500) should be used, I suggest a less uncompromising verb.

The sentence has been rephrased to "This observation indicates that both C1 and C2 precede the O state." (line 459)

2. The figure 5A,B has the pore profiles horizontal, typically they are oriented vertically to match the structures.

Fixed now.

Reviewer #2 (Remarks to the Author):

I am mostly satisfied by the answers of the authors.

Thank you. We appreciate it.

However, I do not agree with the title, as it stands: The experiments described do not "reveal" a new mechanism for gating, they refine what has already been described, only it is now in an environment containing lipids.

The title is now changed to "Cryo-EM structures of prokaryotic ligand-gated ion channel GLIC provide insights into gating in a lipid environment".

Also, the authors should be cautious on the fact that other authors just found out that nanodisks can be misleading, and structural results on the same kind of bacterial pLGIC (ELIC) can vary depending on the size of the nanodisks:

<https://pubmed.ncbi.nlm.nih.gov/38167383/>

The authors should discuss this in their Discussion.

In the limitations of this study section the following sentence is included. "Additionally, different scaffold proteins would be tested during the nanodisc preparation to explore their impact on the structural conformation, as indicated by a recent study highlighting the influence of scaffolding proteins on a pLGIC conformation". (line 574)

I would suggest instead a title taken from their Introduction

Cryo-EM structures of nanodisc-reconstituted prokaryotic ligand-gated ion channel GLIC provide insights on the gating mechanism in a lipid environment

The title is now changed to “Cryo-EM structures of prokaryotic ligand-gated ion channel GLIC provide insights into gating in a lipid environment”.

Reviewer #3 (Remarks to the Author):

For the second revision, the team performed new analyses and revised the text, adjusted the figures, and added supplementary structural movies. I appreciate the thoughtful responses and edits from my and other reviewers' comments. Two rounds of review have now passed with major concerns from all three reviewers at each step.

Thank you for your valuable suggestions.

While the manuscript continues to improve, and the structural data are interesting, I still do not support publication of the current version. A major outstanding issue for me is that the writing still needs improvement- condense, remove non-essential qualifying terms, limit acronyms and jargon, increase precision and accuracy, generally be less informal. The title and abstract are examples of being too long, too detailed, without conveying a succinct and crystal clear conceptual advance appropriate for a broadly-read journal. The issues start at the very beginning. The first few lines of the introduction could be loosely interpreted to suggest that pLGICs in bacteria, and GLIC specifically, mediate fast chemical transmission between neurons, which they do not. I am concerned that as written the text will confuse readers or lead them to an incorrect understanding.

The title is now changed to “Cryo-EM structures of prokaryotic ligand-gated ion channel GLIC provide insights into gating in a lipid environment”.

The abstract is now condensed. Several sentences in main texts have been rephrased to improve the precision and accuracy. Confusing sentences are now removed.

I also remain unconvinced about state assignment, as reviewer 1 mentioned. These channels desensitize; that state should be the most stable at low pH. I understand there are often disconnects we must live with between structural biology and physiology, but assigning the low pH 'closed' states to intermediates on the way to activation rather than to the more stable desensitized state(s) does not make intuitive sense to me.

We completely agree with this comment. This goes back to our previous rebuttal where we have referenced Catherine et al. (PMID: 37026398). They raised a question whether GLIC has canonical D-state or not and also questioned of defining the D-state in GLIC. We reiterated the same question and proposed several approaches to further investigate the possible D-state in GLIC.